# METRIC $k$-CLUSTERING USING ONLY WEAK COMPARISON ORACLES

**Rahul Raychaudhury**[1]    **Aryan Esmailpour**[2]    **Sainyam Galhotra**[3]    **Stavros Sintos**[2]
[1]Duke University    [2]University of Illinois Chicago    [3]Cornell University

## ABSTRACT

Clustering is a fundamental primitive in unsupervised learning. However, classical algorithms for $k$-clustering (such as $k$-median and $k$-means) assume access to exact pairwise distances, which is an unrealistic requirement in many modern applications. We study clustering in the *Rank-model (R-model)*, where access to distances is entirely replaced by a *quadruplet oracle* that provides only relative distance comparisons. In practice, such an oracle can represent learned models or human feedback, and is expected to be noisy and entail an access cost.

Given a metric space with $n$ input items, we design randomized algorithms that, using only a noisy quadruplet oracle, compute a set of $O(k \cdot \mathsf{polylog}(n))$ centers along with a mapping from the input items to the centers such that the clustering cost of the mapping is at most constant times the optimum $k$-clustering cost. Our method achieves a query complexity of $O(n \cdot k \cdot \mathsf{polylog}(n))$ for arbitrary metric spaces and improves to $O((n + k^2) \cdot \mathsf{polylog}(n))$ when the underlying metric has bounded doubling dimension. When the metric has bounded doubling dimension we can further improve the approximation from constant to $1 + \varepsilon$, for any arbitrarily small constant $\varepsilon \in (0, 1)$, while preserving the same asymptotic query complexity. Our framework demonstrates how noisy, low-cost oracles, such as those derived from large language models, can be systematically integrated into scalable clustering algorithms.

## 1 INTRODUCTION

Clustering is a fundamental problem in unsupervised learning. Traditional methods like $k$-center, $k$-median, and $k$-means all rely on computing pairwise distances. For their output clusters to be meaningful, these distances must reflect the user's notion of semantic similarity. However, designing such tailored distance measures is especially difficult for complex data like images. Even when a distance function is well defined, evaluating distances between certain types of objects can be prohibitively expensive.

Motivated by these challenges, there has been a long line of work that avoids direct distance computations and instead uses *oracles*. Oracles serve as abstractions for machine learning models or human feedback that provide partial information about the relative distance between the points. Oracle-based models have been studied for $k$-clustering Bateni et al. (2024); Braverman et al. (2025a); Addanki et al. (2021); Galhotra et al. (2024); Raychaudhury et al. (2025), hierarchical clustering Emamjomeh-Zadeh & Kempe (2018); Chatziafratis et al. (2018); Ghoshdastidar et al. (2019), correlation clustering Ukkonen (2017); Silwal et al. (2023) among others.

In this paper, we study clustering in the *Rank-model* (R-model), where pairwise distances are inaccessible. Instead, one has access to a noisy quadruplet oracle, a function that, given two pairs of input items $(A, B)$ and $(C, D)$, answers the question: *"Is A closer to B, or is C closer to D?"*. Intuitively, quadruplet queries are easier than direct distance queries because they are inherently local and require only relative comparisons, compared to distances which are global. Quadruplet queries are also more practical than the commonly studied optimal-cluster queries, which must return the correct clusters for the queried points. In practice, a quadruplet oracle can be realized in several ways. A natural option is to leverage a large language model (LLM). For instance, two candidate pairs can be presented within a fixed prompt that specifies the intuitive similarity metric of interest, and the model is then asked to return a categorical judgment. Another option is to use an online embedding service: embeddings are computed for each object individually, and the oracle's decision

is obtained by comparing similarity scores for pairs $(A, B)$ and $(C, D)$, returning whichever pair appears more similar. A further possibility is to train a dedicated quadruplet oracle using learning-to-rank methods on annotated data Liu et al. (2009), where the labels themselves may come from crowdsourcing. Regardless of the implementation, we generally expect the oracle to be noisy and to have some cost associated with access. For example, with embedding-based oracles, accuracy depends on how well the embedding space aligns with the semantic notion of similarity. In terms of access costs, LLMs and embedding services incur a direct financial cost.

The study of clustering in the R-model was initiated by Addanki et al. (2021), who considered problems such as $k$-center and hierarchical clustering. Subsequently, Galhotra et al. (2024) showed that no $o(n)$-approximation, where $n$ is the number of items in the metric space, is possible for $k$-median and $k$-means clustering without distance information and introduced the Rank-Measure (RM) model: Along with a quadruplet oracle, they allow access to a distance oracle that returns the exact distance between two input items. They further established several results in this setting. Recently, Raychaudhury et al. (2025), showed that $k$-clustering is possible in the RM-model using $O(nk\,\mathrm{polylog}\,n)$ quadruplet queries and only $O(\mathrm{polylog}\,n)$ distance queries. When the doubling dimension of the input metric space is bounded, they further improve the quadruplet queries to $O((n + k^2)\mathrm{polylog}\,n)$ while distance queries remain $O(\mathrm{polylog}\,n)$. These query complexities are near-optimal within logarithmic factors.

Although these results provide strong guarantees, two challenges remain. First, in practice, a strong distance oracle may just not be available to evaluate distances accurately. Second, it is critical in prior work to interleave distance queries with quadruplet queries, which can be problematic. For example, obtaining exact distances may itself require solving an NP-hard problem, creating a computational bottleneck. Motivated by these considerations, in this paper, we ask, what is the best we can do for clustering when we have access only to a noisy quadruplet oracle? In Appendix A, we show that at least $2k - 1$ centers are necessary to obtain any $o(n)$-approximation algorithm for $k$-median/means clustering in the R-model. Hence, we ask the following questions:

*"In the R-model, can we compute a set of $O(k\,\mathrm{polylog}\,n)$ centers and a mapping from each item to a center, using $O(n\,k\,\mathrm{polylog}\,n)$ quadruplet queries, such that the clustering cost is comparable to the optimal cost with $k$ centers?"*

In many practical settings, the data live in high dimensions but have low intrinsic complexity (e.g., small doubling dimension) Nakis et al. (2025); Roweis & Saul (2000); Tenenbaum et al. (2000). For example, Euclidean space with a fixed number of dimensions is a metric space with a constant doubling dimension. A natural question is whether this additional structure may be beneficial. Specifically, we ask:

*"When the intrinsic dimensionality is small, can we reduce the query complexity to $O(n\,\mathrm{polylog}\,n)$? Moreover, can we improve the approximation quality?"*

In this paper, we provide affirmative answers to both questions. We emphasize that our algorithms do not directly output exactly $k$ centers. Instead, they return a set of $O(k\,\mathrm{polylog}\,n)$ centers together with an assignment of every input point to one of these centers, yielding a clustering whose cost is within a constant factor of the optimal $k$-clustering cost. We note that having such a small set of centers along with a mapping function is very useful in practice, as it shifts the burden of clustering to a substantially smaller set. For instance, consider clustering MNIST digits $0, \ldots, 9$. If one can extract a representative subset of only a few hundred images along with a good mapping, then human annotators would only need to identify the correct class of these, while the mapping automatically ensures that the remaining images are mapped to the correct cluster. Next, we present the formal setting and summarize our contributions.

## 1.1 PROBLEM SETUP AND CONTRIBUTIONS

We require some preliminary definitions before formally presenting the model and our results. Let $\Sigma = (\mathsf{V}, d)$ be a finite metric space with $d : \mathsf{V} \times \mathsf{V} \to \mathbb{R}_{\geq 0}$. We consider metric spaces with $|\mathsf{V}| = n$. Any such space can be viewed as a weighted complete graph. We use $\mathcal{E}$ to denote the set of all edges between vertices in $\mathsf{V}$. The *doubling dimension* $\dim(\Sigma)$ is the smallest $\delta$ such that every ball of radius $\rho$ can be covered by $2^{\delta}$ balls of radius $\rho/2$. We say $\Sigma$ has bounded doubling dimension if $\dim(\Sigma) \leq \delta_0$ for some fixed constant $\delta_0$.

**Clustering Cost.** For $v \in V$ and $U \subseteq V$, define $d(v, U) = \min_{u \in U} d(v, u)$. Let $k, p \in \mathbb{Z}_{\geq 1}$. For $U, W \subseteq V$, let $\text{COST}_U^p(W) = \sum_{w \in W} (d(w, U))^p$. For $W \subseteq V$, the optimal $(k, p)$-clustering cost is $\text{OPT}_k^p(W) = \min_{U \subseteq V, |U|=k} \text{COST}_U^p(W)$; if $W = V$, we simply write $\text{OPT}_k^p$. A $\beta$-approximation algorithm for $(k, p)$ clustering returns a set $\mathcal{A}$ such that $\text{COST}_{\mathcal{A}}^p(V) \leq \beta \cdot \text{OPT}_k^p$, where $\beta \geq 1$. For $p = 1, 2$ the $(k, p)$ clustering corresponds to $k$-median and $k$-means clustering, respectively. Throughout the paper we assume that $p = O(1)$.

**Coresets and Coreset+.** Next, we recall the standard definition of a coreset and introduce the stronger notion of a Coreset+.

$O(1)$-coreset: A set $T \subseteq V$ along with a weight function $\omega : T \to \mathbb{R}_{>0}$ is called an $O(1)$-coreset for $(k, p)$-clustering if any subset $\mathcal{A} \subseteq T$ such that $|\mathcal{A}| = k$ and $\sum_{t \in T} \omega(t)(d(t, \mathcal{A}))^p \leq \beta \cdot \min_{U \subseteq T, |U|=k} \sum_{t \in T} \omega(t)(d(t, U))^p$ satisfies $\text{COST}_{\mathcal{A}}^p(V) \leq O(\beta) \cdot \text{OPT}_k^p$.

$(k, \varepsilon)$-coreset: For a real number $\varepsilon \in (0, 1)$, a set $T \subseteq V$ along with a weight function $\omega : V \to \mathbb{R}_{>0}$ is called a $(k, \varepsilon)$-coreset for $(k, p)$-clustering if for any subset $\mathcal{A} \subseteq V$ with $|\mathcal{A}| = k$, it holds that $(1 - \varepsilon)\text{COST}_{\mathcal{A}}^p(V) \leq \sum_{t \in T} \omega(t)(d(t, \mathcal{A}))^p \leq (1 + \varepsilon)\text{COST}_{\mathcal{A}}^p(V)$.

$\alpha$-*Coreset+*: For a positive real number $\alpha$, a set $C \subseteq V$ together with a mapping $\mathcal{M} : V \to C$ is called an $\alpha$-Coreset+ if $\sum_{v \in V}(d(v, \mathcal{M}(v)))^p \leq \alpha \cdot \text{OPT}_k^p$.

It follows from well-known results (e.g., Har-Peled & Mazumdar (2004); Chen (2009)) that any $O(1)$-Coreset+ directly yields an $O(1)$-coreset: define the weight function $\omega : C \to \mathbb{R}_{>0}$ by $\omega(c) = \sum_{v \in V} \mathbf{1}\{\mathcal{M}(v) = c\}$. Furthermore, for $p = 1$, it follows that an $\varepsilon$-Coreset+ can be converted to a $(k, \varepsilon)$-coreset in the same way.

**R-model.** In the R-model, direct access to the distance function $d$ is unavailable. Instead, all distance-related information must be obtained through a noisy quadruplet oracle. Formally, for an error parameter $\varphi \in \left[0, \frac{1}{4}\right)$, a *quadruplet oracle with probabilistic noise* $\varphi$ is a function $\tilde{\mathcal{Q}} : \mathcal{E} \times \mathcal{E} \to \{\text{YES}, \text{NO}\}$ that, given two edges $\mathbf{e}_1, \mathbf{e}_2 \in \mathcal{E}$, where $\mathbf{e}_1 = \{v_1, v_2\}$, and $\mathbf{e}_2 = \{v_3, v_4\}$, outputs

$$\tilde{\mathcal{Q}}(\mathbf{e}_1, \mathbf{e}_2) = \begin{cases} \text{YES, with probability at least } 1 - \varphi, & \text{if } d(v_1, v_2) \leq d(v_3, v_4), \\ \text{NO, with probability at least } 1 - \varphi, & \text{if } d(v_1, v_2) > d(v_3, v_4). \end{cases}$$

In other words, the oracle fails to identify the closer pair with probability at most $\varphi$. Furthermore, the randomness is independent across distinct queries and is fixed once per edge pair; thus, repeated calls to $\tilde{\mathcal{Q}}(\mathbf{e}_1, \mathbf{e}_2)$ always return the same result, and flipping the order of the edges always flips the answer. This property is referred to as *persistence*.

**Our Results.** As a warm-up, in Appendix A, we show that at least $2k - 1$ centers are necessary to obtain any $o(n)$-approximation algorithm for $k$-median/means clustering in the R-model. On the other hand, we obtain the following positive results for clustering in the R-model.

| | | General | | Doubling | |
|---|---|---|---|---|---|
| | | centers | queries | centers | queries |
| RM-model | Galhotra et al. (2024) | $k$ | $\tilde{O}(nk), \tilde{O}(k^2)$ | $k$ | $\tilde{O}(nk), \tilde{O}(k^2)$ |
| | Raychaudhury et al. (2025) | $k$ | $\tilde{O}(nk), \tilde{O}(1)$ | $k$ | $\tilde{O}(k^2 + n), \tilde{O}(1)$ |
| R-model | Addanki et al. (2021) | $k$ | $\tilde{O}(nk^2)$ | $k$ | $\tilde{O}(nk^2)$ |
| | **NEW** | $\tilde{O}(k)$ | $\tilde{O}(nk)$ | $\tilde{O}(k)$ | $\tilde{O}(k^2 + n)$ |

Table 1: Comparison of our algorithms for $k$-clustering in the R-model with known clustering algorithms in the R-model and RM-model on general and doubling metric spaces. They are all $O(1)$-approximation algorithms. For every algorithm we show the number of centers it returns and the number of oracle queries it executes. In the RM-model, the first (resp. second) quantity in the column queries shows the number of queries to the quadruplet (resp. distance) oracle. The notation $\tilde{O}(\cdot)$ hides a polylog $(n)$ factor. Galhotra et al. (2024); Raychaudhury et al. (2025) studied $k$-median/means clustering. Addanki et al. (2021) only studied the $k$-center clustering in the R-model however their algorithm only holds if the optimal clusters are of size $\Omega(\sqrt{n})$.

- For general metric spaces, we give an algorithm that constructs an $O(1)$-Coreset+ for $(k, p)$-clustering of size $O(k \text{polylog } n)$, using $O(nk \text{polylog } n)$ queries to the quadruplet oracle.

• For metric spaces with bounded doubling dimension (for example, the Euclidean space with constant number of dimensions), we design an algorithm that constructs an $O(1)$-**Coreset+** of size $O(k \text{polylog } n)$, using only $O((n + k^2)\text{polylog } n)$ quadruplet queries.

• For the special case of $k$-median in doubling metrics, we further obtain an $\varepsilon$-**Coreset+** of size $O(k \text{polylog } n)$ with the same query complexity, i.e., $O((n + k^2)\text{polylog } n)$.

Our main results and the comparison with other known algorithms in the RM and R-model are shown in Table 1.

### 1.2 RELATED WORK

Due to the large body of related work, we focus mostly on clustering problems under related oracle-based models. We discuss additional related work in Appendix E.

Addanki et al. (2021) initiated the study of $k$-center clustering with access to a quadruplet oracle. In the R-model (with probabilistic noise), they developed an algorithm under structural assumptions on the optimal clustering. To the best of our knowledge, this remains the only prior work on clustering under purely the R-model.

More recently, Galhotra et al. (2024) showed that even with a perfect quadruplet oracle, no $O(1)$-approximation is possible for $k$-means or $k$-median without distance queries. This motivated their weak-strong framework: a weak oracle provides inexpensive quadruplet comparisons, while a strong oracle supplies exact distances at a higher cost. Within this framework, they designed $O(1)$-approximation algorithms for $k$-center, $k$-median, and $k$-means in general metric spaces, achieving query complexities of $O(nk \text{polylog } n)$ to the quadruplet oracle and $O(k^2 \text{polylog } n)$ to the distance oracle. In a follow-up work, Raychaudhury et al. (2025) improved the number of calls to the distance oracle from $O(k^2 \text{polylog } n)$ to $O(\text{polylog } n)$. They also study clustering problems in metric spaces with bounded doubling dimension, designing $O(1)$-approximation algorithms with $O((n + k^2)\text{polylog } n)$ calls to the quadruplet oracle and $O(\text{polylog } n)$ calls to the distance oracle. Specifically, for $k$-center clustering Raychaudhury et al. (2025) construct $O(1)$-coreset of size $O(k \text{polylog } n)$ executing $O(nk \text{polylog } n)$ (resp. $O((n + k^2)\text{polylog } n)$ in doubling metrics) queries to the quadruplet oracle (R-model). However, using their techniques, no coreset construction for $k$-median or $k$-means can be constructed in the R-model, i.e., an exact distance oracle is necessary.

Independently, Bateni et al. (2024) investigated $k$-clustering and the MST problem in general metrics under a related weak-strong framework. Their strong oracle matches that of Galhotra et al. (2024), but their weak oracle differs: given $u, v \in \mathsf{V}$, it outputs $d(u, v)$ with probability at least $1 - \varepsilon$, and an arbitrary value otherwise. They obtained $O(1)$-approximations for $k$-center, $k$-median, and $k$-means using $O(nk \text{polylog } n)$ weak queries and $O(k^2 \text{polylog } n)$ strong queries. Under the same strong–weak model, Braverman et al. (2025a) studied coresets for clustering problems.

There is a rich line of work on approximate sorting with a probabilistic comparison oracle. In this model, it is well-known that the *maximum dislocation* cannot be improved beyond $O(\log n)$; in particular, no algorithm can guarantee that every element is placed within $o(\log n)$ positions of its true rank. After a series of results Braverman & Mossel (2008); Braverman et al. (2016); Geissmann et al. (2017; 2020), it was recently shown in Geissmann et al. (2025) that $O(\log n)$ dislocation can be achieved with $O(n \log n)$ queries with high probability, when the noise $\varphi < 1/4$.

## 2 TECHNICAL PRELIMINARIES

Let $\Sigma = (\mathsf{V}, d)$ be a metric space with $d : \mathsf{V} \times \mathsf{V} \to \mathbb{R}_{\geq 0}$. We consider finite metric spaces with $|\mathsf{V}| = n$. Any finite metric space can be viewed as a weighted complete graph, and we often use graph terminology. For $\mathsf{U}, \mathsf{W} \subseteq \mathsf{V}$, let $\mathcal{E}(\mathsf{U}, \mathsf{W}) = \{\{u, w\} \mid u \in \mathsf{U}, w \in \mathsf{W}, u \neq w\}$, and $\mathcal{E}(\mathsf{U}) := \mathcal{E}(\mathsf{U}, \mathsf{U})$. For an edge set $\mathfrak{X}$, let $\mathsf{V}(\mathfrak{X})$ denote the set of vertices incident to edges in $\mathfrak{X}$. For $\mathbf{e} = \{u, v\} \in \mathcal{E}(\mathsf{V})$, we define $d(\mathbf{e}) := d(u, v)$. For $\mathsf{U}, \mathsf{W} \subseteq \mathsf{V}$ and $u \in \mathsf{U}$, define $\text{POS}_\mathsf{W}(u; \mathsf{U}) = 1 + |\{x \in \mathsf{U} : d(x, \mathsf{W}) < d(u, \mathsf{W})\}|$, the position of $u$ when $\mathsf{U}$ is ordered by nearest-neighbor distance to $\mathsf{W}$.

While in the R-model we do not have access to distances, the quadruplet oracle allows us to approximately order edges based on their weights (distance). For an edge set $\mathfrak{X} \subseteq \mathcal{E}$, let $\pi_\mathfrak{X}$ denote an *ordered sequence* of the edges in $\mathfrak{X}$. For an edge $\mathbf{e} \in \mathfrak{X}$, we use $\text{RANK}_\mathfrak{X}(\mathbf{e})$ to denote its position

among the edges in $\mathcal{X}$ when sorted in ascending order of distance, [1] and $\text{RANK}_{\pi_\mathcal{X}}(\mathbf{e})$ for its index in $\pi_\mathcal{X}$. The *dislocation* of $\mathbf{e}$ under $\pi_\mathcal{X}$ is defined as $|\text{RANK}_{\pi_\mathcal{X}}(\mathbf{e}) - \text{RANK}_\mathcal{X}(\mathbf{e})|$. The maximum dislocation of $\pi_\mathcal{X}$ is bounded by $\mathcal{D}$ if the dislocation of every edge in $\mathcal{X}$ is bounded by $\mathcal{D}$. It is easy to verify that if $\pi_\mathcal{X}$ has maximum dislocation $\mathcal{D}$, then any subsequence $\pi \sqsubseteq \pi_\mathcal{X}$ also has maximum dislocation at most $\mathcal{D}$. The next lemma follows from a straightforward application of a result by Geissmann et al. (2025).

**Lemma 2.1.** *Let* $\Sigma = (\mathsf{V}, d)$ *be a metric with* $|\mathsf{V}| = n$, *and* $\mathcal{E} = \mathcal{E}(\mathsf{V})$ *be its edge set. Suppose* $\tilde{\mathcal{Q}} : \mathcal{E} \times \mathcal{E} \to \{\text{YES}, \text{NO}\}$ *is a probabilistic quadruplet oracle with noise* $\varphi \in [0, \frac{1}{4}]$. *There exists an algorithm* PROBSORT *such that for any* $\mathcal{X} \subseteq \mathcal{E}$, *with probability* $1 - n^{-4/3}$, PROBSORT$(\mathcal{X})$ *outputs an ordering* $\pi_\mathcal{X}$ *with maximum dislocation* $O(\log n)$ *using* $O(\max(|\mathcal{X}|, n) \log n)$ *queries to* $\tilde{\mathcal{Q}}$.

A small bound on the maximum dislocation of $\pi_\mathcal{X}$ ensures that the ranks are approximately preserved; however, it offers no guarantees about the relative magnitudes of edges that appear in the wrong order. For such guarantees, we require a stronger notion of approximate ordering. For an index $i \in [|\mathcal{X}|]$, let $\pi_\mathcal{X}[i] \in \mathcal{X}$ denote the edge in the $i$-th position of $\pi_\mathcal{X}$. For a constant $\alpha \geq 1$, we say $\pi_\mathcal{X}$ is $\alpha$-*sorted*, if for all $i < j \in [|\mathcal{X}|]$, $d(\pi_\mathcal{X}[i]) \leq \alpha \, d(\pi_\mathcal{X}[j])$. While in this paper we study quadruplet oracles under the probabilistic noise model, prior work (see, for example Addanki et al. (2021)) also considered a weaker *adversarial noise model*.

*Quadruplet Oracle with Adversarial Noise.*[2] Let $\mu \in \mathbb{R}_{\geq 0}$ be a constant. A quadruplet oracle with *adversarial noise* $\mu$ is a function $\mathcal{Q} : \mathcal{E} \times \mathcal{E} \to \{\text{YES}, \text{NO}\}$ that, given two edges $\mathbf{e}_1 = \{\mathsf{v}_1, \mathsf{v}_2\}$ and $\mathbf{e}_2 = \{\mathsf{v}_3, \mathsf{v}_4\}$, outputs YES if $d(\mathsf{v}_1, \mathsf{v}_2) \leq \frac{1}{1+\mu} d(\mathsf{v}_3, \mathsf{v}_4)$, NO if $d(\mathsf{v}_1, \mathsf{v}_2) \geq (1 + \mu) d(\mathsf{v}_3, \mathsf{v}_4)$, and an adversarially chosen (non-adaptive) answer whenever the ratio $d(\mathsf{v}_1, \mathsf{v}_2)/d(\mathsf{v}_3, \mathsf{v}_4)$ lies in the interval $[1/(1 + \mu), \, 1 + \mu]$.

In other words, the oracle gives the correct response when the edge weights are not relatively close but may be adversarially wrong otherwise. The next lemma shows that under the adversarial noise model, it is possible to compute an $O(1)$-sorted sequence of edges using the quadruplet oracle.

**Lemma 2.2.** *Let* $\Sigma = (\mathsf{V}, d)$ *be a metric with* $|\mathsf{V}| = n$, $\mathcal{E} = \mathcal{E}(\mathsf{V})$ *the edge set, and* $\mathcal{Q}$ *an adversarial quadruplet oracle with noise* $\mu \geq 0$. *There exists an algorithm* ADVSORT *such that for any* $\mathcal{X} \subseteq \mathcal{E}$ *of size* $m$, *with probability* $1 - n^{-4}$, ADVSORT$(\mathcal{X})$ *outputs a* $(1 + \mu)^2$-*sorted sequence* $\pi_\mathcal{X}$ *using* $O(m \, \mathsf{polylog} \, n)$ *queries to* $\mathcal{Q}$.

The proof of Lemma 2.2 can be found in Raychaudhury et al. (2025) and is based on a similar result by Acharya et al. (2018). The actual algorithm, ADVSORT, is a slight modification of the classic randomized QUICKSORT algorithm.

Although in this paper we work in the probabilistic noise model, in the analysis of our algorithms, we show that when certain very specific structural conditions hold, it is feasible to emulate an adversarial quadruplet oracle by appropriate calls to the probabilistic quadruplet oracle. In such situations, we will be able to plug in an *emulated adversarial quadruplet oracle* into ADVSORT to obtain an $O(1)$-sorted ordering of edges. We discuss more in the technical overview.

## 3 TECHNICAL OVERVIEW

Let $\Sigma = (\mathsf{V}, d)$ be a metric space accessible in the R-model, i.e., there exists a noisy probabilistic quadruplet oracle $\tilde{\mathcal{Q}}$ that compares edge weights in $\Sigma$. We assume that the error rate of $\tilde{\mathcal{Q}}$ satisfies $\varphi < 1/4$. Our goal is to design an algorithm which, given parameters $k, p \in \mathbb{Z}_{\geq 1}$ and access to $\tilde{\mathcal{Q}}$, returns a Coreset+ for $(k, p)$-clustering of size $O(k \, \mathsf{polylog} \, n)$. For simplicity, we focus on the $k$-median objective ($p = 1$), but our approach extends naturally to any constant $p > 1$.

The remainder of this section is organized as follows. We first describe a generic approach for computing a Coreset+ under a perfect quadruplet oracle. We then present our algorithm for general metric spaces, ALG-G, which adapts this high-level strategy to the noisy setting using $O(n \, k \, \mathsf{polylog} \, n)$ queries. Next, we introduce ALG-D, which further reduces the query complexity to $O((n + k^2) \, \mathsf{polylog} \, n)$ when $\Sigma$ has bounded doubling dimension. Finally, we outline a refinement method, ALG-DI, which takes a Coreset+ returned by the previous algorithm and builds an

---

[1]For simplicity, we assume all pairwise distances are unique.

[2]Most of our results extend to the adversarial model, but we focus on the more challenging probabilistic case.

$\varepsilon$-Coreset+ with $O(n \operatorname{polylog} n)$ additional queries. Due to space constraints, full details and proofs of these algorithms are deferred to Appendix B, C, and D, respectively.

Let $\mathsf{C}^\star$ denote an optimal $k$-median solution, i.e., $|\mathsf{C}^\star| = k$ and $\operatorname{COST}^1_{\mathsf{C}^\star}(\mathsf{V}) = \operatorname{OPT}^1_k$. Recall that a Coreset+ is defined as a set $\mathsf{C} \subseteq \mathsf{V}$ together with a mapping $\mathcal{M} : \mathsf{V} \to \mathsf{C}$ such that $\sum_{\mathsf{v} \in \mathsf{V}} d(\mathsf{v}, \mathsf{C}) \le O(1) \cdot \operatorname{OPT}^1_k$ for all $\mathsf{v} \in \mathsf{V}$. The generic algorithm is based on a well-known *sampling property* of $k$-median clustering Har-Peled & Mazumdar (2004); Mettu & Plaxton (2002). Suppose we take a random sample $\mathcal{S} \subseteq \mathsf{V}$ of size $\Theta(k \operatorname{polylog} n)$. We call a vertex $\mathsf{v} \in \mathsf{V}$ *good* if $d(\mathsf{v}, \mathcal{S}) < 2d(\mathsf{v}, \mathsf{C}^\star)$, and *bad* otherwise. Let $\mathsf{V}_b \subseteq \mathsf{V}$ denote the set of bad vertices. It can be shown that, with high probability, $|\mathsf{V}_b| = o(|\mathsf{V}|)$.

**Generic Algorithm.** The above sampling property leads to a natural recursive sampling algorithm. Sample a set $\mathcal{S} \subseteq \mathsf{V}$ of $O(k \operatorname{polylog} n)$ vertices, order the vertices in $\mathsf{V}$ in ascending order by their nearest-neighbor distance to $\mathcal{S}$, remove a constant fraction subset of the first half from $\mathsf{V}$, and recurse. The process continues until there are $O(k \operatorname{polylog} n)$ remaining vertices. The union of all samples across rounds (along with the remaining vertices in the last round) constructs an $O(1)$-Coreset+ of size $O(k \operatorname{polylog} n)$. The mapping is defined by assigning each vertex $v \in \mathsf{V}$ to its nearest neighbor among the sampled vertices from the round in which $v$ was removed. The main argument is that in any round, there are sufficiently many good vertices in the second half (with larger distances from $\mathcal{S}$) that can account for the accidentally removed bad vertices. By a careful analysis, one can show that across rounds, there exists a bijection between the bad vertices and these good vertices.

At a high level, our algorithms ALG-G and ALG-D emulate this approach. However, there are several obstacles. In order to succeed, in each round we need to map vertices in $\mathsf{V}$ to (approximate) nearest neighbors in $\mathcal{S}$ before ordering them by distance. Although in the R-model we can order the edges $\mathcal{E}(\mathcal{S}, \mathsf{V})$ with $O(\log n)$ dislocation using the PROBSORT primitive, such an ordering is not sufficient to find approximate nearest neighbors. We need a more sophisticated approach.

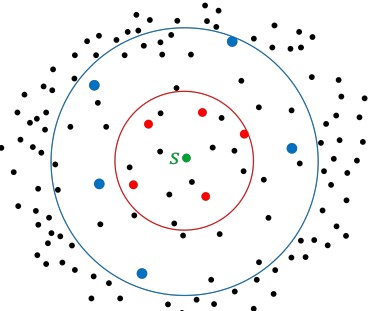

**Overview of ALG-G (Appendix B).** The algorithm operates in rounds. Let $\mathsf{V}_i \subseteq \mathsf{V}$ denote the set of active vertices in round $i$. In each round, the algorithm takes two random samples $\mathcal{S}_i^{(1)}$ and $\mathcal{S}_i^{(2)}$ of sizes $\Theta(k \log^2 n)$ and $\Theta(k \log^3 n)$ respectively. The first sample $\mathcal{S}_i^{(1)}$ plays the same role as the sample set in the generic algorithm described above. The second sample set $\mathcal{S}_i^{(2)}$ is used as follows. The algorithm applies the PROBSORT primitive to approximately order the edges $\mathfrak{X}_i = \mathcal{E}(\mathcal{S}_i^{(1)}, \mathcal{S}_i^{(2)})$, requiring $O(k^2 \operatorname{polylog} n)$ quadruplet queries. From this ordering, the algorithm identifies for each $\mathsf{s} \in \mathcal{S}_i^{(1)}$ two disjoint sets of $\Theta(\log n)$ vertices: a *kernel set* KERNEL$_i(\mathsf{s}) \subset \mathcal{S}_i^{(2)}$ and a *guard set* GUARD$_i(\mathsf{s}) \subset \mathcal{S}_i^{(2)}$. These sets satisfy the following properties:

(i) For every $\mathsf{s} \in \mathcal{S}_i^{(1)}$ and every $\mathsf{v} \in$ KERNEL$_i(\mathsf{s}) \cup$ GUARD$_i(\mathsf{s})$, POS$_\mathsf{s}(\mathsf{v}, \mathsf{V}_i) \le \frac{|\mathsf{V}_i|}{k \operatorname{polylog} n}$.

(ii) For every $\mathsf{s} \in \mathcal{S}_i^{(1)}$, any $\mathsf{w} \in$ KERNEL$_i(\mathsf{s})$, and any $\mathsf{g} \in$ GUARD$_i(\mathsf{s})$, $d(\mathsf{s}, \mathsf{w}) < d(\mathsf{s}, \mathsf{g})$.

The first property implies that for any $\mathsf{s} \in \mathcal{S}_i^{(1)}$ vertices in both

Figure 1: Let $\mathsf{s} \in \mathcal{S}_i^{(1)}$ be the green vertex. The vertices in KERNEL$_i(\mathsf{s})$ are shown in red, and those in GUARD$_i(\mathsf{s})$ are shown in blue. All remaining vertices in $\mathsf{V}$ are depicted as black points. The combined set KERNEL$_i(\mathsf{s}) \cup$ GUARD$_i(\mathsf{s})$ consists of vertices that are close to $\mathsf{s}$ in rank relative to all vertices in $\mathsf{V}$, with those in KERNEL$_i(\mathsf{s})$ being closer to $\mathsf{s}$ than those in GUARD$_i(\mathsf{s})$. All black vertices inside the red circle will be filtered out. No vertex outside the blue circle will be filtered out. Some vertices between the red and blue circles may be filtered out.

KERNEL$_i(\mathsf{s})$ and GUARD$_i(\mathsf{s})$ are very close to $\mathsf{s}$ in terms of rank. The second property ensures that kernel vertices are strictly closer than their respective guard vertices. An example of a kernel and guard set is shown in Figure 1. These sets play complementary roles: guard vertices are used to filter all vertices from $\mathsf{V}_i$ that are too close to $\mathcal{S}_i^{(1)}$, while kernel vertices are used to compute approximate nearest neighbors in $\mathcal{S}_i^{(1)}$ for the rest of the vertices.

*Filtering.* Next, the algorithm for each $\mathsf{v} \in \mathsf{V}_i \setminus (\mathcal{S}_i^{(1)} \cup \mathcal{S}_i^{(2)})$ and $\mathsf{s} \in \mathcal{S}_i^{(1)}$, compares against the guard vertices to compute *proximity scores*:

$$\text{PCOUNT}_{\mathsf{s}}(\mathsf{v}) := \sum_{\mathsf{g} \in \text{GUARD}_i(\mathsf{s})} \mathbf{1}\{\tilde{\mathcal{Q}}(\{\mathsf{s},\mathsf{v}\}, \{\mathsf{s},\mathsf{g}\}) \text{ returns } "d(\mathsf{s},\mathsf{v}) \leq d(\mathsf{s},\mathsf{g})"\}, \tag{1}$$

Based on these scores, the algorithm computes a subset $\mathsf{V}'_i = \{\mathsf{v} \in \mathsf{V}_i \setminus (\mathcal{S}_i^{(1)} \cup \mathcal{S}_i^{(2)}) \mid \max_{\mathsf{s} \in \mathcal{S}_i^{(1)}} \text{PCOUNT}_{\mathsf{s}}(\mathsf{v}) < \lfloor m_{\mathsf{win}}/2 \rfloor\}$, where $m_{\mathsf{win}}$ is a suitable threshold of size $\Theta(\log n)$. Computing $\mathsf{V}'_i$ requires $O(nk\,\text{polylog}\,n)$ quadruplet queries. Recall that the generic algorithm, in each round, orders vertices based on their nearest neighbor distance to the sample and removes a fraction of the first half. In the presence of noise, we cannot find the nearest neighbors of all vertices in $\mathsf{V}_i$. It turns out that $\mathsf{V}'_i$ has sufficient structural properties for the algorithm to be able to find their nearest neighbors in $\mathcal{S}_i^{(1)}$. In the analysis, we show that filtering guarantees that no vertex in $\mathsf{V}'_i$ is closer to a sample vertex $\mathsf{s} \in \mathcal{S}_i^{(1)}$ than the kernel vertices $\text{KERNEL}_i(\mathsf{s})$, while at the same time not discarding too many additional vertices. Formally, filtering ensures that with high probability, $|\mathsf{V}'_i| \geq \frac{3}{5}|\mathsf{V}_i|$ and $\forall \mathsf{v} \in \mathsf{V}'_i, \ \forall \mathsf{s} \in \mathcal{S}_i^{(1)} : \ d(\mathsf{v},\mathsf{s}) > r_{\mathsf{s}}$, where $r_{\mathsf{s}} := \max_{\mathsf{w} \in \text{KERNEL}_i(\mathsf{s})} d(\mathsf{s},\mathsf{w})$ denotes the *kernel radius* of $\mathsf{s}$. An example of filtering vertices is shown in Figure 1.

*Finding approximate nearest neighbors.* The key insight is the following. Consider any two sample vertices $\mathsf{s}_1, \mathsf{s}_2 \in \mathcal{S}_i^{(1)}$ and any vertices $\mathsf{v}_1, \mathsf{v}_2 \in \mathsf{V} \setminus \mathcal{S}_i^{(2)}$, and suppose the following conditions hold:

(i) $d(\mathsf{s}_1, \mathsf{v}_1) > r_{\mathsf{s}_1}$ and $d(\mathsf{s}_2, \mathsf{v}_2) > r_{\mathsf{s}_2}$,

(ii) we know which kernel has the smaller radius, i.e., whether $r_{\mathsf{s}_1} \leq r_{\mathsf{s}_2}$ or vice versa.

In the analysis, we show that when the above conditions hold, by making appropriate comparisons with the smaller kernel, we can design a test procedure that answers whether $d(\mathsf{s}_1, \mathsf{v}_1) < d(\mathsf{s}_2, \mathsf{v}_2)$. The test is correct with high probability when the two distances differ by more than a 2 factor, i.e., $d(\mathsf{s}_1, \mathsf{v}_1)/d(\mathsf{s}_2, \mathsf{v}_2) > 2$ or $d(\mathsf{s}_2, \mathsf{v}_2)/d(\mathsf{s}_1, \mathsf{v}_1) > 2$. Observe that this behavior is similar to a *quadruplet oracle with adversarial noise* with error $\mu = 1$ (see Section 2). We design a procedure ALG-TESTER (see Section B.2), based on this test.

Our algorithm runs ADVSORT (see Section 2) with ALG-TESTER as the comparator to order the edges in $\mathcal{Y}_i = \mathcal{E}(\mathcal{S}_i^{(1)}, \mathsf{V}'_i)$. Whenever ALG-TESTER is asked to compare two edges from $\mathcal{Y}_i$, the first condition is satisfied by the construction of $\mathsf{V}'_i$, and ALG-TESTER uses the ordering of $\mathcal{X}_i$ to determine which kernel has a smaller radius. By the preceding discussion, on all such queries, it behaves akin to an adversarial quadruplet oracle with noise $\mu = 1$. This immediately yields a 4-approximate nearest neighbor in $\mathcal{S}_i^{(1)}$ for every vertex in $\mathsf{V}'_i$. Each call to ALG-TESTER triggers $O(\log n)$ quadruplet queries, and ADVSORT calls ALG-TESTER a total of $O(nk\,\text{polylog}\,n)$ times.

The previous step computes an approximate nearest neighbor for each vertex in $\mathsf{V}'_i$ within the set $\mathcal{S}_i^{(1)}$. In the next step, the algorithm again applies ADVSORT, as before, to approximately order the vertices in $\mathsf{V}'_i$ according to their estimated nearest-neighbor distances, and then identifies a prefix $\mathsf{V}''_i \subseteq \mathsf{V}'_i$. Intuitively, this set $\mathsf{V}''_i$ corresponds to the set of vertices removed by the generic algorithm in a round. The algorithm also *maps* each vertex in $\mathsf{V}''_i$ to their neighbor in $\mathcal{S}_i^{(1)}$ found previously. Next, the algorithm recurses on $\mathsf{V}_{i+1} = \mathsf{V}_i \setminus (\mathsf{V}''_i \cup \mathcal{S}_i^1 \cup \mathcal{S}_i^2)$. The process terminates after $r = O(\log n)$ rounds, at which point we output the set $\mathsf{C} = \bigcup_{i=1}^r (\mathcal{S}_i^{(1)} \cup \mathcal{S}_i^{(2)})$ of $O(k\,\text{polylog}\,n)$ centers that consists of the union of all samples. The mapping function $\mathcal{M}$ is defined by the per-round mappings of the sets $\mathsf{V}''_i$ to the sets $\mathcal{S}_i^{(1)}$. The complete algorithm along with the analysis is available in Appendix B. We conclude with the next theorem.

**Theorem 3.1.** *Let $\Sigma = (\mathsf{V}, d)$ be a finite metric space of size $|\mathsf{V}| = n$, which is accessible under the R-model. There exists a randomized algorithm ALG-G that, given parameters $k, p \in \mathbb{Z}_+$, with high probability returns an $O(1)$-Coreset+ for $(k, p)$-clustering of size $O(k\,\text{polylog}\,n)$ using $O(nk\,\text{polylog}\,n)$ calls to the quadruplet oracle.*

**Overview of ALG-D (Appendix C).** When $\Sigma$ has bounded doubling dimension, we present an algorithm ALG-D (Section C.1) that reduces the total number of quadruplet queries to $O((n + k^2)\,\text{polylog}\,n)$. The algorithm follows a similar recursive-sampling framework as ALG-G. Each round begins by drawing two random samples $\mathcal{S}_i^{(1)}$ and $\mathcal{S}_i^{(2)}$, and computing the kernel and guard sets for every $\mathsf{s} \in \mathcal{S}_i^{(1)}$. However, in order to achieve the desired query complexity bounds, we cannot afford to perform the full filtering and nearest-neighbor procedures used in ALG-G. Instead, the algorithm proceeds as follows.

*Partitioning.* We first partition the vertices in $\mathcal{S}_i^{(1)}$ into classes $\mathcal{S}_i^{(1,1)}, \ldots, \mathcal{S}_i^{(1,\chi_i)}$ such that no two vertices in the same class are *close*. To do this, we construct a *conflict graph* $G_i$ whose vertex set is $\mathcal{S}_i^{(1)}$, and add an edge between any two vertices that are close. Closeness is determined using proximity scores derived from the guard sets, as in Equation (1). In the analysis, we show that $G_i$ is $O(\log n)$-*degenerate*, i.e., every subgraph of $G_i$ has a vertex of degree at most $O(\log n)$. It is known that a $\xi$-degenerate graph can be properly colored with $\xi + 1$ colors using a simple greedy algorithm Lick & White (1970). We use such a coloring to obtain the classes $\mathcal{S}_i^{(1,1)}, \ldots, \mathcal{S}_i^{(1,\chi_i)}$.

*Nearest neighbors.* Our next goal is to compute approximate nearest neighbors for vertices in $\mathsf{V}_i \setminus (\mathcal{S}_i^{(1)} \cup \mathcal{S}_i^{(2)})$ with respect to each class $\mathcal{S}_i^{(1,j)}$. In Raychaudhury et al. (2025) it was shown that given two disjoint sets $\mathsf{U}, \mathsf{W} \subseteq \mathsf{V}$ and access to an adversarial quadruplet oracle with noise $\mu$, that can answer quadruplet queries of the form $(\{\mathbf{e}_1\}, \{\mathbf{e}_2\})$, where $\mathbf{e}_1 \in \mathcal{E}(\mathsf{U}, \mathsf{U})$, $\mathbf{e}_2 \in \mathcal{E}(\mathsf{U}, \mathsf{W})$, one can construct a data structure, such that given a vertex $\mathsf{v} \in \mathsf{W}$, it returns a subset of size $O(\text{polylog } n)$ containing at least one vertex $\mathsf{u} \in \mathsf{U}$, such that $d(\mathsf{w}, \mathsf{u}) \leq O(1) \cdot d(\mathsf{w}, \mathsf{U})$. Our plan is to apply this result to each class $\mathcal{S}_i^{(1,j)}$ for each $j = 1, \ldots, \chi_i$, where we set $\mathsf{U} = \mathcal{S}_i^{(1,j)}$ and $\mathsf{W} = \mathsf{V}_i \setminus (\mathcal{S}_i^{(1)} \cup \mathcal{S}_i^{(2)})$. To simulate the adversarial oracle, we again use ALG-TESTER as in the general metric case. However, unlike in ALG-G, we have not pre-filtered close vertices in $\mathsf{V}_i \setminus (\mathcal{S}_i^{(1)} \cup \mathcal{S}_i^{(1)})$. If we apply ALG-TESTER to edges containing such vertices, it is not guaranteed to behave like an adversarial quadruplet oracle. Hence, we adopt a *lazy filtering* strategy: whenever ALG-TESTER is invoked to compare a pair of edges, we first compute proximity scores to ensure that the vertices are not too close. If a vertex is found to be too close, we discard it. This ensures correctness while avoiding the heavy global filtering step of ALG-G. In the analysis, we show that the overall number of quadruplet oracles required in this step is $O(n \text{ polylog } n)$.

Post finding approximate nearest neighbors, we proceed similarly as in ALG-G. The full details are in Appendix C. Overall, we get the next result.

**Theorem 3.2.** *Let $\Sigma = (\mathsf{V}, d)$ be a finite metric space of size $|\mathsf{V}| = n$ with bounded doubling dimension, which is accessible under the R-model. There exists a randomized algorithm* ALG-D *that, given parameters $k, p \in \mathbb{Z}_+$, with high probability returns an $O(1)$-**Coreset+** for $(k, p)$-clustering of size $O(k \text{ polylog } n)$ using $O((n + k^2) \text{ polylog } n)$ calls to the quadruplet oracle.*

**Overview of ALG-DI (Appendix D).** When the underlying metric has bounded doubling dimension, we show that, given a Coreset+ consisting of a set $\mathsf{C} \subseteq \mathsf{V}$ and a mapping $\mathcal{M} : \mathsf{V} \to \mathsf{C}$ computed via ALG-G or ALG-D, we can compute an $\varepsilon$-Coreset+, $(\mathsf{C}^+, \mathcal{M}^+)$ using only $O(n \text{ polylog } n)$ additional queries, for $k$-median clustering.
Consider a vertex $\mathsf{s} \in \mathsf{C}$, and let $\mathsf{U}_\mathsf{s}$ denote the set of vertices mapped to it, i.e., $\mathsf{U}_\mathsf{s} = \{\mathsf{u} \in \mathsf{V} : \mathcal{M}(\mathsf{u}) = \mathsf{s}\}$. Let $\alpha_\mathsf{s} = \max_{\mathsf{u} \in \mathsf{U}_\mathsf{s}} d(\mathsf{u}, \mathsf{s})$. Suppose we order the vertices in $\mathsf{U}_\mathsf{s}$ by $d(\cdot, \mathsf{s})$ and partition them into buckets: $B_0$ has maximum distance $\alpha_\mathsf{s}/n^2$, $B_1$ covers $(\alpha_\mathsf{s}/n^2, 2\alpha_\mathsf{s}/n^2]$, $B_2$ covers $(2\alpha_\mathsf{s}/n^2, 4\alpha_\mathsf{s}/n^2]$ and so on, doubling the outer radius each time. It is easy to see that there are at most $O(\log n)$ buckets since $\alpha_\mathsf{s} \leq O(1) \cdot \mathsf{OPT}_k^1$. Since the doubling dimension is bounded, it is well known that if we had access to such a partition, we could improve the approximation quality by choosing a constant number of vertices from each bucket (see, e.g., Har-Peled & Mazumdar (2004)). Our high-level strategy is to follow a similar approach. Unfortunately, without access to distances, we cannot compute such a partition directly. Our algorithm operates as follows.

For every $\mathsf{s} \in \mathsf{C}$, it first obtains a $O(1)$-sorted ordering of the edges $\mathcal{E}(\mathsf{s}, \mathsf{U}_\mathsf{s})$. This does not require any quadruplet queries and can simply be extracted from the post-ANN ordering computed by ALG-D or ALG-G. We then perform a multistep sampling on this order: in the first round, we sample $\Theta(\text{polylog } n)$ vertices from $\mathsf{U}_\mathsf{s}$, in the second round from the last half, then from the last quarter, and so on, halving the suffix each time. Let $\mathsf{W}_\mathsf{s}$ be all the samples obtained from $\mathsf{U}_\mathsf{s}$. Repeating this for all $\mathsf{s} \in \mathsf{C}$ accumulates $O(k \text{ polylog } n)$ new samples. We set $\mathsf{C}^+ = \mathsf{C} \cup (\bigcup_{\mathsf{s} \in \mathsf{C}} \mathsf{W}_\mathsf{s})$. The algorithm then orders the set $\bigcup_{\mathsf{s} \in \mathsf{C}} \mathcal{E}(\mathsf{W}_\mathsf{s}, \mathsf{U}_\mathsf{s})$ using PROBSORT and uses that information to construct a new mapping $\mathcal{M}^+$; we skip those details here. This step uses $O(n \text{ polylog } n)$ additional queries.

The crux is the analysis. Although we cannot ensure that we hit every bucket for a $\mathsf{s} \in \mathsf{C}$ exactly, we show that our sampling hits relevant distance scales with high probability. We carefully argue that the revised mapping is indeed sufficient.

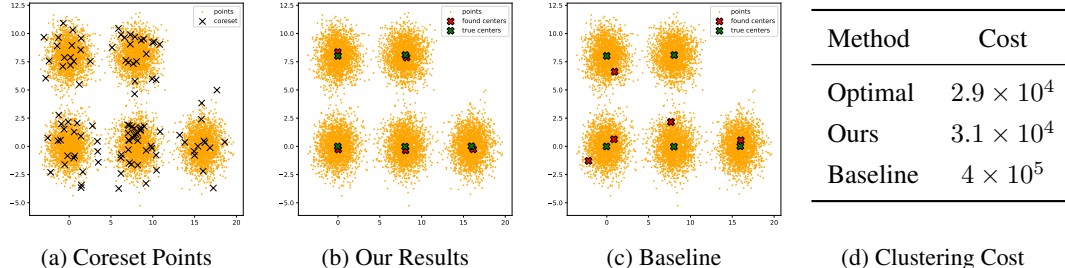

| | (a) Coreset Points | (b) Our Results | (c) Baseline | (d) Clustering Cost |

Figure 2: Comparison of the $k$-means clustering results obtained by our method against the baseline and the optimal clustering.

**Theorem 3.3.** *Let $\Sigma = (\mathsf{V}, d)$ be a finite metric space of size $|\mathsf{V}| = n$ with bounded doubling dimension, which is accessible under the R-model. There exists a randomized algorithm* ALG-DI *that, given a parameter $k \in \mathbb{Z}_+$, with high probability returns an $\varepsilon$-**Coreset+** for $k$-median clustering of size $O(k\mathsf{polylog}\, n)$ using $O((n + k^2)\, \mathsf{polylog}\, n)$ calls to the quadruplet oracle, where $\varepsilon \in (0,1)$ is an arbitrarily small constant.*

## 4 EXPERIMENTS

In this section, we present preliminary experimental results based on a basic implementation of our algorithm for $k$-means clustering. We evaluate the quality of the results obtained from our algorithm, comparing it against the clustering obtained by $k$-means++ algorithm over the ground truth data points, and the clustering obtained by a baseline that always trusts the answers of the noisy quadruplet oracle. The complete descriptions of our algorithm and the baseline are provided later. We refer to the cluster centers obtained from running $k$-means++ algorithm over the ground truth data as *true centers* and the $k$-means cost of the true centers as optimal cost[3]. We use both a synthetic and two real datasets whose points are used as the ground truth data in the experiments. By default, we simulate the quadruplet oracle with probabilistic noise and set the error rate to $\varphi = 0.15$. The oracle has access to the ground truth data and answers correctly with probability $1 - \varphi = 0.85$. Both the baseline algorithm and our algorithm can only access the data via this noisy oracle and do not have direct access to the ground truth data. The main goal of our experiments is to show the effectiveness of our **Coreset+** construction and especially the mapping of the points to the **Coreset+** centers. All methods were implemented in Python, and the experiments were conducted using the free version of Google Colab.

### 4.1 EXPERIMENTAL SETUP

**Synthetic dataset.** For our experiments, we generate a synthetic dataset consisting of two-dimensional points arranged in $k = 5$ clusters of approximately equal size. For each cluster, the points are randomly generated following the Gaussian distribution with a standard deviation equal to 1. After generating the data, all the points are randomly shuffled to avoid any relation between ordering and the clusters. Using this procedure, we generate a dataset of $10^4$ synthetic points as shown in Figure 2. We use this dataset as the ground truth data in the experiments.

**Real datasets.** We use two real-world datasets used to evaluate clustering algorithms ( Braverman et al. (2025a); Huang et al. (2019)): the *Adult* dataset Becker & Kohavi (1996) and the *Default of Credit Card Clients* dataset Yeh (2009). For Adult dataset, we use eight numerical attributes, resulting in a collection of roughly 50,000 points in 8 dimensions. For Credit dataset, we select nine numerical attributes, resulting in a collection of roughly 30,000 points in 9 dimensions. The values of all attributes are then normalized to be in the range $[0, 1]$. Similarly to Braverman et al. (2025a), due to computational constraints, we sample 2,000 data points from each dataset using Meyerson sampling Meyerson (2001): We begin with an empty set $S$ and process the points sequentially. The first point is added to $S$. For each subsequent point, we add it to $S$ with probability proportional to its distance from the current set $S$. We employ Meyerson sampling instead of uniform sampling, as its distance-based selection better preserves the geometric structure of the dataset and reduces the risk of underrepresenting smaller clusters.

---

[3]$k$-means clustering is an NP-hard problem, and thus the $k$-means++ algorithm does not always yield the optimal solution. Nevertheless, we treat the clustering derived from the ground-truth data points as optimal.

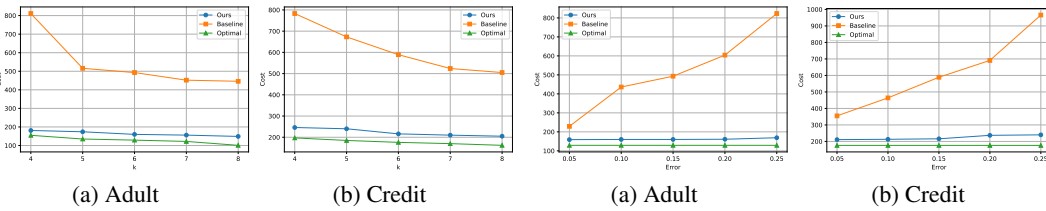

| (a) Adult | (b) Credit | (a) Adult | (b) Credit |

Figure 3: Clustering cost varying $k$       Figure 4: Clustering cost varying $\varphi$

**Our approach.** We first run ALG-G from Theorem 3.1 to obtain an $O(1)$-Coreset+. Then, as described in Section 1, the obtained centers and the mapping are used to construct an $O(1)$-coreset: each Coreset+ point gets a weight equal to the number of points mapped to it. After constructing the weighted coreset, we assume access to a distance oracle, as in the RM-model of Raychaudhury et al. (2025). We then apply $k$-means++ algorithm on the weighted coreset to obtain the final set of $k$ centers, invoking the distance oracle when necessary.

**Baseline.** The baseline directly applies the Generic algorithm (Section 3) using the quadruplet oracle. Although the oracle may return an incorrect answer with probability $\varphi$, the algorithm assumes every response is correct and proceeds accordingly. From the returned set of points and the mapping, we construct a weighted set of centers (coreset) in the same manner as our algorithm: each sampled center is assigned a weight equal to the number of data points mapped to it. Finally, we run $k$-means++ algorithm on the weighted coreset to obtain the final $k$ centers, invoking the distance oracle whenever the exact distance between two coreset points is required.

## 4.2 EXPERIMENTAL RESULTS

**Results on synthetic dataset.** We observe that the coreset produced by our algorithm contains only 187 points. This corresponds to a $98\%$ reduction in size, from the original $10^4$ input points down to just 187 coreset points. Figure 2a shows the set of coreset points. Instead of using the original large dataset, this much smaller coreset is used to obtain the final $k$ centers running the $k$-means++ algorithm, assuming access to a distance oracle, as described above.

We next evaluate the clustering quality by comparing the results of our algorithm against both the baseline and the ground-truth centers. As shown in Figure 2, the centers identified by our method closely match the true centers (see also Figure 2b). By contrast, the baseline fails to recover some clusters due to erroneous mapping, as illustrated in Figure 2c, leading to noticeably poorer clustering performance. This limitation arises because the baseline unconditionally trusts the noisy oracle during the Generic algorithm. Table 2d reports the corresponding clustering costs. The $k$-means cost is defined as the sum of squared distances from each point to its mapped center. Our method achieves a cost within $7\%$ of the optimal solution, while the baseline incurs a cost exceeding the optimum by more than $1200\%$.

**Results on real datasets.** Next, we compare the clustering cost of our method with both the baseline and the optimal cost on real datasets. In Figure 3, we fix the error rate at $\varphi = 0.15$ and vary the number of clusters $k \in \{4, 5, 6, 7, 8\}$. In both datasets, our method achieves a clustering cost that is very close to the optimum. In contrast, the baseline consistently yields a clustering cost that is 2.5 to 4 times higher than that of our approach. As expected, the clustering cost of all methods decreases as the number of clusters increases. Finally, in Figure 4, we fix the number of clusters at $k = 6$ and vary the oracle's probabilistic noise level $\varphi \in \{0.05, 0.1, 0.15, 0.2, 0.25\}$. As before, in both datasets, the clustering cost achieved by our method remains close to the optimum. Moreover, consistent with our theoretical guarantees, the clustering cost of our method is essentially independent of $\varphi$. In contrast, the clustering cost of the baseline increases substantially as $\varphi$ grows.

## 5 CONCLUSION

We proposed near-optimal algorithms for constructing coresets for $(k, p)$-clustering in the R-model. Our results open several directions for future research. First, it is interesting to study whether the techniques can be extended to other clustering objectives (such as hierarchical clustering or sum-of-radii clustering) and to related graph problems (such as the Minimum Spanning Tree problem) in the R-model. Second, we plan to generalize our framework to non-metric graphs, to alternative oracle models (such as triplet oracles), and to different error models.

ACKNOWLEDGMENTS

This work has been partially supported by NSF grants IIS-2348919, IIS-2402823, and a grant by Infosys.

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

## A    LOWER BOUND FOR $k$-CLUSTERING IN THE R-MODEL.

**Theorem A.1.** *Every $o(n)$-approximation algorithm for the $k$-median/means clustering problem in the R-model must contain at least $2k - 1$ centers, where $n$ is the number of vertices in the input metric space and $3 \le k = O(1)$.*

*Proof.* We show the proof for the $k$-median clustering, however all results can be extended to $k$-means clustering. We show a stronger result, assuming that the probabilistic noise is 0, i.e., perfect oracle.

We construct a finite metric space $\Sigma = (\mathsf{V}, d)$ of $n$ vertices as follows. For simplicity, we assume that $n' = \frac{n}{k}$ is an integer. A set of $k - 1$ vertices $U = u_1, \ldots, u_{k-1}$ are contained in $\mathsf{V}$ with $d(u_i, u_j) = \zeta$, for every $i \ne j \in [k - 1]$, for a parameter $\zeta > 0$ that we specify later. Furthermore, there exists a set $Y$ of $n'$ vertices $y_1^{(1)}, \ldots, y_{n'}^{(1)}$ such that $d(y_i^{(1)}, y_j^{(1)}) = 0$ for every $i \ne j \in [n']$. Moreover, there are $k - 1$ groups of vertices, $X_2 = \{x_1^{(2)}, \ldots, x_{n'-1}^{(2)}\}$, $X_3 = \{x_1^{(3)}, \ldots, x_{n'-1}^{(3)}\}$, $\ldots$, $X_k = \{x_1^{(k)}, \ldots, x_{n'-1}^{(k)}\}$ such that each group contains $n' - 1$ vertices, while $d(x_i^{(h)}, x_j^{(h)}) = 0$ for every $h \in \{2, \ldots, k\}$ and every $i \ne j \in [n' - 1]$. Let $X = \bigcup_{i=2,\ldots,k} X_i$. For every $y_i \in Y$ (for $i \in [n']$) and $x \in X$, $d(x, y_i) = 1$. For every $x_1 \in X_i$ and $x_2 \in X_j$, where $i \ne j \in \{2, \ldots, k\}$, $d(x_1, x_2) = 1$. Finally, for every vertex $t \in X \cup Y$ and $u \in U$, $d(t, u) = \zeta$. It is easy to verify that the function $d(\cdot)$ satisfies the triangle inequality, so $\Sigma$ is a metric space.

We note that under the R-model (with probabilistic noise 0) an algorithm cannot distinguish between an instance of the metric space where $\zeta = n^3$ and $\zeta = 2$. Indeed, for both choices, $\zeta > 1$, so the ordering of all pairwise distances of the vertices in $\mathsf{V}$ remain the same for $\zeta = n^3$ and $\zeta = 2$.

Next, we show that no set of centers of size at most $2k - 2$ returns an $o(n)$-approximation solution for both undistinguishable instances of $k$-median clustering on $\mathsf{V}$ with $\zeta = 2$ and $\zeta = n^3$.

First, consider the case where $\zeta = n^3$. The optimum set for $k$-median clustering for $\mathsf{V}$ is $P_1^* = U \cup \{y\}$, where $y \in Y$, with $\mathsf{OPT}^1 = \mathsf{COST}_{P_1^*}^1(\mathsf{V}) = \left(\frac{n}{k} - 1\right)(k - 1)$.

If $\zeta = 2$, then the optimum set for $k$-median clustering on $\mathsf{V}$ is $P_2^* = \{y\} \cup \left(\bigcup_{h=2,\ldots k} x_h\right)$, where $y \in Y$, and $x_h \in X_h$ for every $h \in \{2, \ldots, k\}$ with $\mathsf{OPT}^1 = \mathsf{COST}_{P_2^*}^1(\mathsf{V}) = 2(k - 1)$.

For $\zeta = 2$, any $o(n)$-approximation solution $R \subseteq \mathsf{V}$ of size at most $2k - 2$ must include one (arbitrary) vertex from every group $X_i$ (for every $i = \{2, \ldots, k\}$) and one (arbitrary) vertex from $Y$. If this is not the case, then the $k$-median cost will be at least $\frac{n}{k} - 1$ leading to an approximation ratio $\frac{\frac{n}{k} - 1}{2(k-1)} = \Omega(n)$. Without loss of generality assume that a set $R$ contains one arbitrary vertex from $Y$, one arbitrary vertex for every $X_i$, for every $i = \{2, \ldots, k\}$, and $k - 2$ arbitrary vertices from $U$. Notice that $\mathsf{COST}_R^1(\mathsf{V}) = 2$. Recall that in the R-model no algorithm can distinguish the two instances with $\zeta = 2$ and $\zeta = n^3$. Hence, if $R$ is selected as a solution of size $2k - 2$ for the $k$-median instance when $\zeta = 2$, then $R$ will also be selected as a solution for the $k$-median instance when $\zeta = n^3$. However, when $\zeta = n^3$, then $\mathsf{OPT}^1 = \left(\frac{n}{k} - 1\right)(k - 1)$ and $\mathsf{COST}_R^1(\mathsf{V}) = n^3$ so the approximation ratio is $\Omega(n^2)$. The result follows.

$\square$

## B    ALGORITHM FOR GENERAL METRICS

We are given a set of vertices $\mathsf{V}$ from a metric space $\Sigma = (\mathsf{V}, d)$ with $|\mathsf{V}| = n$, and access to a noisy quadruplet oracle $\tilde{\mathcal{Q}}$ over $\Sigma$ with known error rate $\varphi < \frac{1}{2}$. Given cluster size $k$ and parameter $p \in \mathbb{Z}_{\ge 1} \cup \{\infty\}$, the algorithm outputs a set $\mathsf{C} \subseteq \mathsf{V}$ of size $\tilde{O}(k \operatorname{polylog} n)$ and a mapping $\mathcal{M} : \mathsf{V} \to \mathsf{C}$. For clarity of exposition, we restrict to the $k$-median case ($p = 1$) and assume $\varphi$ is bounded by a fixed constant below $1/4$. The rest of this section is organized as follows: Section B.1 presents the main algorithm, while Section B.2 describes a key subroutine.

## B.1 ALG-G

**Overview.** The algorithm proceeds in rounds. Initially, $V_1 = V$. In round $i$, the algorithm processes $V_i$ and removes a subset of vertices to obtain a smaller set $V_{i+1} \subset V_i$. The algorithm terminates when $|V_i| = O(k \log^3 n)$ or if the number of rounds exceeds $r = O(\log n)$. Throughout, we use $\mathcal{D} = O(\log n)$ to denote the maximum dislocation bound of PROBSORT on any set of edges.

**Round $i$.** The algorithm proceeds as follows.

1. *Sampling.* Sample uniformly at random (with replacement) a set of vertices $\mathcal{S}_i^{(1)}, \mathcal{S}_i^{(2)} \subseteq V_i$ of sizes $m_{\mathsf{S1}} = c_{\mathsf{S1}} k \log^2 n$ and $m_{\mathsf{S2}} = c_{\mathsf{S2}} k \log^3 n$ respectively, where $c_{\mathsf{S1}}, c_{\mathsf{S2}}$ are suitable constants. Let $\mathcal{S}_i := \mathcal{S}_i^{(1)} \cup \mathcal{S}_i^{(2)}$.

2. *Ordering edges.* Let $\mathcal{X}_i = \mathcal{E}(\mathcal{S}_i^{(1)}, \mathcal{S}_i^{(2)})$ and compute $\pi_{\mathcal{X}_i} = \text{PROBSORT}(\mathcal{X}_i)$.

3. *Kernel and guard sets.* Let $m_{\mathsf{win}} = 2\max\{c_{\mathsf{win}} \log n, \mathcal{D}\}$, where $c_{\mathsf{win}}$ is a sufficiently large constant. For each $\mathsf{s} \in \mathcal{S}_i^{(1)}$, let $\mathcal{X}_{i,\mathsf{s}} = \mathcal{E}(\mathsf{s}, \mathcal{S}_i^{(2)})$ and $\pi_{\mathcal{X}_{i,\mathsf{s}}}$ the ordering of $\mathcal{X}_{i,\mathsf{s}}$ induced by $\pi_{\mathcal{X}_i}$. For every $\mathsf{s} \in \mathcal{S}_i^{(1)}$, compute

$$\text{KERNEL}_i(\mathsf{s}) = \{\mathsf{w} \in \mathcal{S}_i^{(2)} : \text{RANK}_{\pi_{\mathcal{X}_{i,\mathsf{s}}}}[\{\mathsf{s}, \mathsf{w}\}] \leq m_{\mathsf{win}}\},$$

$$\text{GUARD}_i(\mathsf{s}) = \{\mathsf{g} \in \mathcal{S}_i^{(2)} : m_{\mathsf{win}} + 2\mathcal{D} < \text{RANK}_{\pi_{\mathcal{X}_{i,\mathsf{s}}}}[\{\mathsf{s}, \mathsf{g}\}] \leq 2m_{\mathsf{win}} + 2\mathcal{D}\}.$$

4. *Filtering.* For any $\mathsf{s} \in \mathcal{S}_i^{(1)}$ and $\mathsf{v} \in V_i$, define the *proximity score*

$$\text{PCOUNT}_\mathsf{s}(\mathsf{v}) := \sum_{\mathsf{g} \in \text{GUARD}_i(\mathsf{s})} \mathbf{1}\{\tilde{\mathcal{Q}}(\{\mathsf{s}, \mathsf{v}\}, \{\mathsf{s}, \mathsf{g}\}) \text{ returns "}d(\mathsf{s}, \mathsf{v}) \leq d(\mathsf{s}, \mathsf{g})\text{"}\},$$

where $\mathbf{1}\{\mathsf{condition}\}$ is the indicator function that returns 1 if the $\mathsf{condition}$ holds, and 0 otherwise. Compute $V_i' = \{\mathsf{v} \in V_i \setminus \mathcal{S}_i \mid \max_{\mathsf{s} \in \mathcal{S}_i^{(1)}} \text{PCOUNT}_\mathsf{s}(\mathsf{v}) < \lfloor m_{\mathsf{win}}/2 \rfloor\}$.

5. *Approximate nearest neighbors.* Let $\mathcal{Y}_i = \mathcal{E}(\mathcal{S}_i^{(1)}, V_i')$. Compute $\pi_{\mathcal{Y}_i} = \text{ADVSORT}(\mathcal{Y}_i)$ using ALG-TESTER (Subsection B.2) as the comparator. For each $\mathsf{v} \in V_i'$, let $\mathsf{f}_\mathsf{v}$ be its *first* incident edge in $\pi_{\mathcal{Y}_i}$. Set $\mathcal{N}_i = \{\mathsf{f}_\mathsf{v} : \mathsf{v} \in V_i'\}$.

6. *Safe-set and mapping.* Let $\pi_{\mathcal{N}_i}$ be the ordering of $\mathcal{N}_i$ induced by $\pi_{\mathcal{Y}_i}$. Define $V_i'' = \{\mathsf{v} \in V_i' : \text{RANK}_{\pi_{\mathcal{N}_i}}(\mathsf{f}_\mathsf{v}) \leq |V_i|/4\}$. For every $\mathsf{v} \in V_i''$, define $\mathcal{M}_i(\mathsf{v})$ as the endpoint of $\mathsf{f}_\mathsf{v}$ in $\mathcal{S}_i^{(1)}$. For every $\mathsf{v} \in \mathcal{S}_i$, define $\mathcal{M}_i(\mathsf{v}) := \mathsf{v}$.

7. *Recurse.* Set $V_{i+1} = V_i \setminus (V_i'' \cup \mathcal{S}_i)$ and proceed to next round.

**Final output.** Let $i^\star$ denote the last round. The final centers are $\mathsf{C} := V_{i^\star} \cup \bigcup_{i=1}^{i^\star - 1} \mathcal{S}_i$. Let $\mathcal{M}_{i^\star}$ denote the function that maps every vertex in $V_{i^\star}$ to itself. Finally, define the global mapping $\mathcal{M}$ as the union of the per round mappings $\mathcal{M} := \bigcup_{i=1}^{i^\star} \mathcal{M}_i$. This is a function from $V \mapsto \mathsf{C}$ since each $\mathcal{M}_i$ is defined on a disjoint round-specific set.

**Weighted coreset.** We can also obtain a weighted coreset by assigning to each $\mathsf{u} \in \mathsf{C}$ a weight equal to the number of original vertices mapped to it, namely $w(\mathsf{u}) := |\{\mathsf{v} \in V \mid \mathcal{M}(\mathsf{v}) = \mathsf{u}\}|$. The pair $(\mathsf{C}, w)$ defines the coreset.

## B.2 ALG-TESTER

Whenever the tester is invoked in round $i$, it is given two edges $\mathsf{e}_1 = \{\mathsf{s}_1, \mathsf{v}_1\}$ and $\mathsf{e}_2 = \{\mathsf{s}_2, \mathsf{v}_2\}$, with $\mathsf{s}_1, \mathsf{s}_2 \in \mathcal{S}_i^{(1)}$ and $\mathsf{v}_1, \mathsf{v}_2 \in V_i \setminus \mathcal{S}_i^{(2)}$. It is also given access to the kernel sets $\text{KERNEL}_i(\mathsf{s}_1), \text{KERNEL}_i(\mathsf{s}_2)$ and the global ordering $\pi_{\mathcal{X}_i}$. From these, it forms $\mathcal{Z} = \mathcal{E}(\mathsf{s}_1, \text{KERNEL}_i(\mathsf{s}_1)) \cup \mathcal{E}(\mathsf{s}_2, \text{KERNEL}_i(\mathsf{s}_2))$, and computes $\pi_{\mathcal{Z}}$, the ordering of $\mathcal{Z}$ induced by $\pi_{\mathcal{X}_i}$.

1. **Case $\mathsf{s}_1 \neq \mathsf{s}_2$.**

(a) *Kernel Selection.* Let $\mathbf{e}^\star$ be the last edge in $\pi_{\mathcal{Z}}$. Determine whether it belongs to $\mathcal{E}(\mathsf{s}_1, \text{KERNEL}_i(\mathsf{s}_1))$ or $\mathcal{E}(\mathsf{s}_2, \text{KERNEL}_i(\mathsf{s}_2))$. Without loss of generality, assume $\mathbf{e}^\star \in \mathcal{E}(\mathsf{s}_1, \text{KERNEL}_i(\mathsf{s}_1))$. Remove every vertex $\mathsf{w} \in \text{KERNEL}_i(\mathsf{s}_2)$ such that $\text{RANK}_{\pi_{\mathcal{Z}}}(\{\mathsf{s}_2, \mathsf{w}\}) \in [2m_{\mathsf{win}} - \mathcal{D}, \, 2m_{\mathsf{win}}]$, and call the remainder $\text{KERNEL}'(\mathsf{s}_2)$.

(b) *Majority Test.* Compute

$$\text{TCOUNT} = \sum_{\mathsf{w} \in \text{KERNEL}'(\mathsf{s}_2)} \mathbf{1}\{\tilde{\mathcal{Q}}(\{\mathsf{s}_1, \mathsf{v}_1\}, \{\mathsf{w}, \mathsf{v}_2\}) \text{ says } "d(\mathsf{s}_1, \mathsf{v}_1) > d(\mathsf{w}, \mathsf{v}_2)"\}.$$

Output "$d(\mathsf{s}_1, \mathsf{v}_1) > d(\mathsf{s}_2, \mathsf{v}_2)$" if $\text{TCOUNT} > \lfloor \frac{1}{2}|\text{KERNEL}'(\mathsf{s}_2)|\rfloor$, and the opposite otherwise.

2. **Case $\mathsf{s}_1 = \mathsf{s}_2$.** Set $\text{KERNEL}'(\mathsf{s}_2) := \text{KERNEL}_i(\mathsf{s}_2)$ and run the same *majority test* as in Case (a).

## B.3 ANALYSIS

We first establish that no quadruplet query is ever repeated between rounds and within certain steps of a round. We refer to this as the *isolation property*. Next, in Section B.3.1 we prove certain properties that hold in each round. Finally, in Section B.3.2, we combine the guarantees to establish global correctness.

**Lemma B.1** (Isolation). *The algorithm satisfies the following properties:*

1. *No two rounds rely on the outcome of the same quadruplet query.*

2. *Within any round, there is no overlap between the quadruplet queries used by* $\text{PROBSORT}(\mathcal{X}_i)$, *with those required to compute any proximity score* $\text{PCOUNT}_\mathsf{s}(\cdot)$, *or those required by any invocation of* ALG-TESTER.

*Proof.* Since $\mathsf{V}_{i+1} = \mathsf{V}_i'' \setminus (\mathcal{S}_i^{(1)} \cup \mathcal{S}_i^{(2)})$, it follows that $\mathsf{V}_i \cap (\mathcal{S}_{i-1}^{(1)} \cup \mathcal{S}_{i-1}^{(2)}) = \emptyset$. Therefore, the sampled sets $\mathcal{S}_i^{(1)}, \mathcal{S}_i^{(2)} \subseteq \mathsf{V}_i$ are disjoint from previous samples. This implies that the corresponding $\mathcal{X}_i = \mathcal{E}(\mathcal{S}_i^{(1)}, \mathcal{S}_i^{(2)})$, $\text{KERNEL}_i(\cdot) \subseteq \mathcal{S}_i^{(2)}$, and $\text{GUARD}_i(\cdot) \subseteq \mathcal{S}_i^{(2)}$ are disjoint from previous rounds. Hence, no quadruplet query is ever reused across rounds.

Within a round $i$, for any $\mathsf{s} \in \mathcal{S}_i^{(1)}$, the computation of $\text{PCOUNT}_\mathsf{s}(\mathsf{v})$ is always performed with $\mathsf{v} \in \mathsf{V}_i \setminus \mathcal{S}_i^{(2)}$. Thus, every query triggered by it involves one edge from $\mathcal{E}(\mathcal{S}_i^{(1)}, \mathsf{V}_i \setminus \mathcal{S}_i^{(2)})$. Similarly, every query issued by ALG-TESTER includes at least one edge from $\mathcal{E}(\mathcal{S}_i^{(1)}, \mathsf{V}_i \setminus \mathcal{S}_i^{(2)})$. Since $\mathcal{X}_i \cap \mathcal{E}(\mathcal{S}_i^{(1)}, \mathsf{V}_i \setminus \mathcal{S}_i^{(2)}) = \emptyset$, quadruplet queries used for computing proximity scores or those used by ALG-TESTER cannot overlap with any query used by $\text{PROBSORT}(\mathcal{X}_i)$, which always compares two edges from $\mathcal{X}_i$.

$\square$

### B.3.1 PER-ROUND GUARANTEES

We first show that in every round the ordering $\pi_{\mathcal{X}_i}$ has bounded dislocation, the kernel and guard sets are confined to the top-ranked vertices, and that kernels are always closer than guards

**Lemma B.2.** *In any round $i$, with probability at least $1 - n^{-\Omega(1)}$ the following properties hold simultaneously:*

(i) $\pi_{\mathcal{X}_i}$ *has maximum dislocation $\mathcal{D} = O(\log n)$.*

(ii) *For every $\mathsf{s} \in \mathcal{S}_i^{(1)}$ and every $\mathsf{v} \in \text{KERNEL}_i(\mathsf{s}) \cup \text{GUARD}_i(\mathsf{s})$, $\text{POS}_\mathsf{s}(\mathsf{v}, \mathsf{V}_i) \leq \frac{|\mathsf{V}_i|}{100 m_{\mathsf{s}1}}$.*

(iii) *For every $\mathsf{s} \in \mathcal{S}_i^{(1)}$, any $\mathsf{w} \in \text{KERNEL}_i(\mathsf{s})$, and any $\mathsf{g} \in \text{GUARD}_i(\mathsf{s})$, $d(\mathsf{s}, \mathsf{w}) < d(\mathsf{s}, \mathsf{g})$.*

*Proof.* We first start with the straightforward proof of (i). By Lemma B.1, quadruplet queries are not repeated between rounds. Therefore, applying Lemma 2.1 to $\mathcal{X}_i = \mathcal{E}(\mathcal{S}_i^{(1)}, \mathcal{S}_i^{(2)})$, we have that $\pi_{\mathcal{X}_i}$ has maximum dislocation at most $\mathcal{D}$ with probability at least $1 - n^{-4/3}$.

Then we show (ii). Fix some round $i$. For any $\mathsf{s} \in \mathcal{S}_i^{(1)}$, define $B(\mathsf{s}) := \left\{\mathsf{v} \in \mathsf{V}_i : \text{POS}_\mathsf{s}(\mathsf{v}, \mathsf{V}_i) \leq \frac{|\mathsf{V}_i|}{100\, m_{\mathsf{S1}}}\right\}$. For any fixed $\mathsf{s} \in \mathcal{S}_i^{(1)}$, since $\mathcal{S}_i^{(2)}$ is drawn uniformly and independently with replacement, the random variable $N_\mathsf{s} := \left|\mathcal{S}_i^{(2)} \cap B(\mathsf{s})\right|$ is binomial with mean $\mathbb{E}[N_\mathsf{s}] = m_{\mathsf{S2}} \cdot \frac{|B(\mathsf{s})|}{|\mathsf{V}_i|} = \frac{m_{\mathsf{S2}}}{100\, m_{\mathsf{S1}}}$. Recall that $m_{\mathsf{S2}} = c_{\mathsf{S2}}\, k \log^3 n$ and $m_{\mathsf{S1}} = c_{\mathsf{S1}} k \log^2 n$. Thus $\mathbb{E}[N_\mathsf{s}] = \frac{c_{\mathsf{S2}} \log n}{c_{\mathsf{S1}}}$. Observe that $m_{\mathsf{win}} = 2\max\{c_{\mathsf{win}} \log n, \mathcal{D}\} = \Theta(\log n)$.

Assume $c_{\mathsf{S2}}$ is large enough so that $\mathbb{E}[N_\mathsf{s}] \geq \max\{1000 m_{\mathsf{win}}, 1000 \log n\}$. By a standard Chernoff bound, we get

$$\Pr\left[N_\mathsf{s} < 500 m_{\mathsf{win}}\right] \leq \Pr\left[N_\mathsf{s} < \tfrac{1}{2}\mathbb{E}[N_\mathsf{s}]\right] \leq \exp(-\mathbb{E}[N_\mathsf{s}]/8) \leq n^{-6}.$$

Taking a union bound over all $\mathsf{s} \in \mathcal{S}_i^{(1)}$ with probability at least $1 - n^{-4}$, $N_\mathsf{s} \geq 500 m_{\mathsf{win}}$ for every $\mathsf{s} \in \mathcal{S}_i^{(1)}$.

So far the lower bound on $N_\mathsf{s}$ and (i) hold with probability $1 - n^{-\Omega(1)}$. We now continue the analysis conditioned on this. Fix some $\mathsf{s} \in \mathcal{S}_i^{(1)}$. Since any restriction $\pi_{\mathcal{X}_{i,\mathsf{s}}}$ inherits the same maximum dislocation bound as $\pi_{\mathcal{X}_i}$, it also has maximum dislocation $\mathcal{D}$. Thus, for any edge $\{\mathsf{s}, \mathsf{s}'\} \in \mathcal{E}(\mathsf{s}, \mathcal{S}_i^{(2)})$, if $\mathsf{s}' \notin B(\mathsf{s})$, then $\text{RANK}_{\pi_{\mathcal{X}_{i,\mathsf{s}}}}[\{\mathsf{s}, \mathsf{s}'\}]$ must be at least $N_\mathsf{s} - \mathcal{D} \geq 500 m_{\mathsf{win}} - \mathcal{D}$. Therefore, for every edge $\{\mathsf{s}, \mathsf{v}\} \in \mathcal{E}(\mathsf{s}, \text{KERNEL}_i(\mathsf{s}) \cup \text{GUARD}_i(\mathsf{s}))$, it holds that $\text{RANK}_{\pi_{\mathcal{X}_{i,\mathsf{s}}}}[\{\mathsf{s}, \mathsf{v}\}] \leq 2 m_{\mathsf{win}} + 2\mathcal{D} \leq 500 m_{\mathsf{win}} - \mathcal{D}$. By definition, $\mathsf{v} \in \mathcal{S}_i^{(2)}$, so $\mathsf{v} \in B(\mathsf{s})$.

Finally, we show (iii). We note that for any $\mathsf{w} \in \text{KERNEL}_i(\mathsf{s})$ and $\mathsf{g} \in \text{GUARD}_i(\mathsf{s})$ we have

$$\text{RANK}_{\pi_{\mathcal{X}_{i,\mathsf{s}}}}\{\mathsf{s}, \mathsf{w}\} \leq m_{\mathsf{win}}, \qquad \text{RANK}_{\pi_{\mathcal{X}_{i,\mathsf{s}}}}\{\mathsf{s}, \mathsf{g}\} > m_{\mathsf{win}} + 2\mathcal{D}.$$

With maximum dislocation $\mathcal{D}$, the true rank of every kernel edge is at most $m_{\mathsf{win}} + \mathcal{D}$ while the true rank of every guard edge exceeds $m_{\mathsf{win}} + \mathcal{D}$. Hence, every kernel edge precedes every guard edge in the true order, and therefore $d(\mathsf{s}, \mathsf{w}) < d(\mathsf{s}, \mathsf{g})$. Thus, all of (i),(ii), (iii) hold.

$\square$

Next, we show that the proximity score is a reliable indicator: vertices nearer than all kernels yield large scores, while those farther than all guards yield small scores.

**Lemma B.3.** *In any round $i$, conditioned on Lemma B.2, with probability at least $1 - n^{-4}$, the following hold simultaneously for every $\mathsf{s} \in \mathcal{S}_i^{(1)}$ and every $\mathsf{v} \in \mathsf{V}_i \setminus \mathcal{S}_i^{(2)}$:*

*(i) If $d(\mathsf{s}, \mathsf{v}) \leq \max_{\mathsf{w} \in \text{KERNEL}_i(\mathsf{s})} d(\mathsf{s}, \mathsf{w})$ then $\text{PCOUNT}_\mathsf{s}(\mathsf{v}) > \lfloor m_{\mathsf{win}}/2 \rfloor$.*

*(ii) If $d(\mathsf{s}, \mathsf{v}) > \max_{\mathsf{g} \in \text{GUARD}_i(\mathsf{s})} d(\mathsf{s}, \mathsf{g})$ then $\text{PCOUNT}_\mathsf{s}(\mathsf{v}) \leq \lfloor m_{\mathsf{win}}/2 \rfloor$.*

*(iii) If $\text{PCOUNT}_\mathsf{s}(\mathsf{v}) > \lfloor m_{\mathsf{win}}/2 \rfloor$ then $\text{POS}_\mathsf{s}(\mathsf{v}, \mathsf{V}_i) \leq \frac{|\mathsf{V}_i|}{100 m_{\mathsf{S1}}}$*

*Proof.* Fix a round $i$ and a vertex $\mathsf{s} \in \mathcal{S}_i^{(1)}$. Consider any $\mathsf{v} \in \mathsf{V}_i \setminus \{\mathsf{s}\}$. Recall that

$$\text{PCOUNT}_\mathsf{s}(\mathsf{v}) := \sum_{\mathsf{g} \in \text{GUARD}_i(\mathsf{s})} \mathbf{1}\{\tilde{\mathcal{Q}}(\{\mathsf{s}, \mathsf{v}\}, \{\mathsf{s}, \mathsf{g}\}) \text{ says } d(\mathsf{s}, \mathsf{v}) \leq d(\mathsf{s}, \mathsf{g})\}.$$

By Lemma B.2, we have $\max_{\mathsf{w} \in \text{KERNEL}_i(\mathsf{s})} d(\mathsf{s}, \mathsf{w}) < \min_{\mathsf{g} \in \text{GUARD}_i(\mathsf{s})} d(\mathsf{s}, \mathsf{g})$. Moreover, by Lemma B.1, quadruplet queries involved in computing $\text{PCOUNT}_\mathsf{s}(\cdot)$ do not overlap with those used by $\text{PROBSORT}(\mathcal{X}_i)$.

If $d(\mathsf{s}, \mathsf{v}) \leq \max_{\mathsf{w} \in \text{KERNEL}_i(\mathsf{s})} d(\mathsf{s}, \mathsf{w})$, then the correct answer to every comparison is "Yes". For sufficiently large $c_{\mathsf{win}}$ in $m_{\mathsf{win}} = 2\max\{c_{\mathsf{win}} \log n, \mathcal{D}\}$, Chernoff bounds yield $\Pr\left[\text{PCOUNT}_\mathsf{s}(\mathsf{v}) < \lfloor m_{\mathsf{win}}/2 \rfloor\right] \leq \exp(-\Theta(m_{\mathsf{win}})) \leq n^{-7}$. Similarly, if $d(\mathsf{s}, \mathsf{v}) > \max_{\mathsf{g} \in \text{GUARD}_i(\mathsf{s})} d(\mathsf{s}, \mathsf{g})$, then the correct answer to every comparison is "No" and $\Pr\left[\text{PCOUNT}_\mathsf{s}(\mathsf{v}) > \lfloor m_{\mathsf{win}}/2 \rfloor\right] \leq n^{-7}$. Taking a union bound over all pairs $(\mathsf{s}, \mathsf{v})$ with $\mathsf{s} \in \mathcal{S}_i^{(1)}$ and $\mathsf{v} \in \mathsf{V}_i$, both events (i) and (ii) hold with probability at least $1 - n^{-4}$.

Finally, from (ii), if $\text{PCOUNT}_\mathsf{s}(\mathsf{v}) > \lfloor m_\mathsf{win}/2 \rfloor$ then $d(\mathsf{s}, \mathsf{v}) \leq \max_{\mathsf{w} \in \text{KERNEL}_i(\mathsf{s})} d(\mathsf{s}, \mathsf{w})$. From Lemma B.2 we have that for every vertex $\mathsf{w} \in \text{KERNEL}_i(\mathsf{s})$ it holds that $\text{POS}_\mathsf{s}(\mathsf{w}, \mathsf{V}_i) \leq \frac{|\mathsf{V}_i|}{100 m_\mathsf{S1}}$. Hence, $\text{POS}_\mathsf{s}(\mathsf{v}, \mathsf{V}_i) \leq \max_{\mathsf{w} \in \text{KERNEL}_i(\mathsf{s})} \text{POS}_\mathsf{s}(\mathsf{w}, \mathsf{V}_i) \leq \frac{|\mathsf{V}_i|}{100 m_\mathsf{S1}}$.

$\square$

Next, we establish that filtering removes only a small fraction of vertices, while ensuring that all survivors are well-separated from the kernel sets.

**Lemma B.4.** *In any round $i$, conditioned on Lemma B.2, with probability $1 - n^{-4}$, the following hold simultaneously:*

(i) *Every $\mathsf{v} \in \mathsf{V}_i'$ satisfies $d(\mathsf{s}, \mathsf{v}) > \max_{\mathsf{w} \in \text{KERNEL}_i(\mathsf{s})} d(\mathsf{s}, \mathsf{w})$ for all $\mathsf{s} \in \mathcal{S}_i^{(1)}$.*

(ii) $|\mathsf{V}_i'| \geq \frac{3}{5}|\mathsf{V}_i|$.

*Proof.* Conditioned on Lemma B.2, the conditions in Lemma B.3 hold with probability at least $1 - n^{-4}$.

Case (i) : Lemma B.3 implies that for every $\mathsf{s} \in \mathcal{S}_i^{(1)}$ and every $\mathsf{v} \in \mathsf{V}_i \setminus \mathcal{S}_i^{(2)}$ if $d(\mathsf{s}, \mathsf{v}) \leq \max_{\mathsf{w} \in \text{KERNEL}_i(\mathsf{s})} d(\mathsf{s}, \mathsf{w})$ then $\text{PCOUNT}_\mathsf{s}(\mathsf{v}) > \lfloor m_\mathsf{win}/2 \rfloor$. By the definition of $\mathsf{V}_i'$, a node $\mathsf{v} \in \mathsf{V}_i \setminus \bar{\mathcal{S}}_i$ is excluded from $\mathsf{V}_i'$ if $\max_{\mathsf{s} \in \mathcal{S}_i^{(1)}} \text{PCOUNT}_\mathsf{s}(\mathsf{v}) \geq \lfloor m_\mathsf{win}/2 \rfloor$. So it follows directly that every $\mathsf{v} \in \mathsf{V}_i'$ must satisfy the claimed inequality for all $\mathsf{s} \in \mathcal{S}_i^{(1)}$.

Case (ii) : Lemma B.2 guarantees that for each $\mathsf{s} \in \mathcal{S}_i^{(1)}$,

$$\text{KERNEL}_i(\mathsf{s}) \cup \text{GUARD}_i(\mathsf{s}) \subseteq \{\mathsf{u} \in \mathsf{V}_i : \text{POS}_\mathsf{s}(\mathsf{u}, \mathsf{V}_i) \leq |\mathsf{V}_i|/(100 m_\mathsf{S1})\}.$$

Lemma B.3 (iii) implies that for every $\mathsf{s} \in \mathcal{S}_i^{(1)}$ and every $\mathsf{v} \in \mathsf{V}_i \setminus \mathcal{S}_i^{(2)}$, if $\text{POS}_\mathsf{s}(\mathsf{v}, \mathsf{V}_i) > |\mathsf{V}_i|/(100 m_\mathsf{S1})$ then $\text{PCOUNT}_\mathsf{s}(\mathsf{v}) \leq \lfloor m_\mathsf{win}/2 \rfloor$. Hence for every vertex $\mathsf{v} \in \mathsf{V}_i \setminus \mathcal{S}_i^{(2)}$ that is not included in $\mathsf{V}_i'$ there exists at least one $\mathsf{s} \in \mathcal{S}_i^{(1)}$ such that $\text{POS}_\mathsf{s}(\mathsf{v}, \mathsf{V}_i) \leq |\mathsf{V}_i|/(100 m_\mathsf{S1})$. Thus, any vertex $\mathsf{s} \in \mathcal{S}_i^{(2)}$ can cause at most $|\mathsf{V}_i|/(100 m_\mathsf{S1})$ vertices from $\mathsf{V}_i \setminus \mathcal{S}_i^{(2)}$ to not be included in $\mathsf{V}_i'$. There are $m_\mathsf{S1}$ samples in $\mathcal{S}_i^{(1)}$ so at most $|\mathsf{V}_i|/100$ vertices will not be included in $\mathsf{V}_i'$. Therefore, $|\mathsf{V}_i'| \geq |\mathsf{V}_i| - |\mathsf{V}_i|/100 \geq (3/5)|\mathsf{V}_i|$, as claimed.

$\square$

**Lemma B.5.** *In any round $i$, conditioned on Lemma B.2, the following holds with probability at least $1 - n^{-4}$: for any query $q = (\{\mathsf{s}_1, \mathsf{v}_1\}, \{\mathsf{s}_2, \mathsf{v}_2\})$ with $\mathsf{s}_1, \mathsf{s}_2 \in \mathcal{S}_i^{(1)}$ and $\mathsf{v}_1, \mathsf{v}_2 \in \mathsf{V}_i \setminus \mathcal{S}_i^{(2)}$,*

*if*

$$d(\mathsf{s}_j, \mathsf{v}_j) > \max_{\mathsf{w} \in \text{KERNEL}_i(\mathsf{s}_j)} d(\mathsf{s}_j, \mathsf{w}) \qquad \text{for } j = 1, 2,$$

*then* $\text{ALG-TESTER}(q)$ *behaves like an adversarial quadruplet oracle with error $\mu = 1$.*

*Proof.* We focus on the case $\mathsf{s}_1 \neq \mathsf{s}_2$; the other case is simpler. Assume that $\mathsf{e}^\star \in \mathcal{E}(\mathsf{s}_1, \text{KERNEL}_i(\mathsf{s}_1))$ such that $\text{RANK}_{\pi_\mathcal{Z}}(\mathsf{e}^\star) = |\mathcal{Z}|$. The tester discards any vertex $\mathsf{u}$ from $\text{KERNEL}_i(\mathsf{s}_2)$ with $\text{RANK}_{\pi_\mathcal{Z}}(\{\mathsf{s}_2, \mathsf{u}\}) \in [|\mathcal{Z}| - \mathcal{D}, |\mathcal{Z}|)$. Since $\pi_\mathcal{Z}$ has maximum dislocation $\mathcal{D}$, every $\mathsf{w} \in \text{KERNEL}_i'(\mathsf{s}_2)$ satisfies

$$d(\mathsf{s}_2, \mathsf{w}) \leq d(\mathsf{e}^\star) < d(\mathsf{s}_1, \mathsf{v}_1).$$

Moreover, assuming the constant $c_\mathsf{win}$ is large enough in $m_\mathsf{win} := 2\max\{c_\mathsf{win} \log n, \mathcal{D}\}$, it holds $|\text{KERNEL}_i'(\mathsf{s}_2)| = 2m_\mathsf{win} - \mathcal{D} = \Omega(\log n)$, since $\mathcal{D} = O(\log n)$.

Each oracle query is correct independently with probability at least $1 - \varphi > 3/4$. Let $m := |\text{KERNEL}_i'(\mathsf{s}_2)|$. The tester outputs "yes" if more than $\tau := \lfloor m/2 \rfloor$ comparisons say $\{\mathsf{s}_1, \mathsf{v}_1\}$ is larger, and "no" otherwise.

*Case 1:* TCOUNT $> \tau$ *(tester outputs "yes").* This can only be wrong if $d(v_2, s_2) > 2d(v_1, s_1)$. In that case, for every $w \in \text{KERNEL}'_i(s_2)$,

$$d(v_2, w) \geq d(v_2, s_2) - d(s_2, w) \geq d(v_2, s_2) - d(e^\star) > 2d(v_1, s_1) - d(e^\star) > d(v_1, s_1).$$

Thus every "yes" response is incorrect, and by Chernoff bounds $\Pr[T > \tau] \leq n^{-8}$.

*Case 2:* TCOUNT $\leq \tau$ *(tester outputs "no").* This can only be wrong if $d(v_1, s_1) > 2d(v_2, s_2)$. In that case, for every $w \in \text{KERNEL}'_i(s_2)$,

$$d(v_2, w) \leq d(v_2, s_2) + d(s_2, w) < 2d(v_2, s_2) < d(v_1, s_1),$$

so every "no" response is incorrect. Again, $\Pr[T \leq \tau] \leq n^{-8}$.

The case $s_1 = s_2$ is analogous: in Case 1 we have directly $d(s_2, w) \leq d(s_2, v_1)$, and in Case 2 we have $d(s_2, w) \leq d(v_2, s_2)$. Thus, in all cases, the tester fails with probability at most $n^{-8}$.

Finally, union bounding over all $O(n^4)$ possible queries in round $i$ gives overall failure probability at most $n^{-4}$, as claimed. □

**Lemma B.6.** *In any round $i$ of* ALG-G*, conditioned on Lemma B.2, with probability at least $1 - n^{-3}$, the following holds:*

*(i) for every $v \in V''_i$, $d(v, \mathcal{M}_i(v)) \leq 4\, d(v, \mathcal{S}^{(1)}_i)$.*

*(ii) $|V''_i| \geq |V_i|/4$.*

*Proof.* Conditioned on Lemma B.2, each of Lemma B.3, Lemma B.4, and Lemma B.5 fails with probability at most $n^{-4}$. By a union bound, all of them hold simultaneously with probability $1 - 3n^{-4}$. We analyze under this assumption.

By Lemma B.4, for every $v \in V'_i$ and every $s \in \mathcal{S}^{(1)}_i$ we have $d(s, v) > \max_{w \in \text{KERNEL}_i(s)} d(s, w)$. Since ADVSORT only calls ALG-TESTER on pairs of edges in $\mathcal{Y}_i = \mathcal{E}(\mathcal{S}^{(1)}_i, V'_i)$, every invocation of ALG-TESTER satisfies the preconditions of Lemma B.5. Specifically, for any query $q = (\{s_1, v_1\}, \{s_2, v_2\})$ with $s_1, s_2 \in \mathcal{S}^{(1)}_i$ and $v_1, v_2 \in V'_i$, the following condition holds for $j = 1, 2$:

$$d(v_j, s_j) > \max_{w \in \text{KERNEL}_i(s_j)} d(s_j, w).$$

Therefore, by Lemma B.5, whenever ALG-TESTER is invoked in behaves like an adversarial quadruplet oracle with error $\mu = 1$. It follows from Lemma 2.2 that the ordering $\pi_{\mathcal{Y}_i}$ is 4-sorted: for any $e_1, e_2 \in \mathcal{Y}_i$,

$$\text{RANK}_{\pi_{\mathcal{Y}_i}}(e_1) \leq \text{RANK}_{\pi_{\mathcal{Y}_i}}(e_2) \Rightarrow d(e_1) \leq 4\, d(e_2).$$

Let $f_v$ be the first edge incident to $v \in V'_i$ in $\pi_{\mathcal{Y}_i}$, and recall $\mathcal{M}_i(v)$ is its endpoint in $\mathcal{S}^{(1)}_i$. Then for any $s \in \mathcal{S}^{(1)}_i$,

$$d(v, \mathcal{M}_i(v)) = d(f_v) \leq 4\, d(\{s, v\}),$$

hence $d(v, \mathcal{M}_i(v)) \leq 4 \cdot d(v, \mathcal{S}^{(1)}_i)$. This proves (i).

For (ii), Lemma B.4 guarantees $|V'_i| \geq \frac{3}{5}|V_i|$. The safe set is

$$V''_i = \{v \in V'_i : \text{RANK}_{\pi_{\mathcal{N}_i}}(f_v) \leq |V_i|/4\}.$$

Since $|V'_i| \geq 3|V_i|/5$, at least $|V_i|/4$ vertices of $V'_i$ satisfy the rank condition, and hence $|V''_i| \geq |V_i|/4$. This proves (ii). □

**Structural Property.** Let $\text{OPT}^1_k(V)$ be the optimal $k$-median cost, and let $C^\star$ denote the set of centers in an optimal solution. Let $\mathcal{L} := \text{OPT}^1_k(V)/n$. For a round $i$, define a vertex $v \in V_i$ as *good* if

$$d(v, \mathcal{S}^{(1)}_i) \leq \max\{\mathcal{L}, 2 \cdot d(v, C^\star)\},$$

and *bad* otherwise. Let $V^g_i, V^b_i$ be the good/bad sets so $V_i = V^g_i \cup V^b_i$.

**Lemma B.7.** *In any round $i$, with probability at least $1 - n^{-3}$,*

$$|\mathsf{V}_i^b| = O\left(\frac{|\mathsf{V}_i|}{\log n}\right).$$

*Proof.* Let $\mathsf{c} \in \mathsf{C}^\star$. Furthermore, let $\mathcal{N}_\mathsf{c} \subseteq \mathsf{V}$ be the set of all vertices that get mapped to $\mathsf{c}$ according to the optimal clustering. Fix a round $i$, and let $\mathsf{s}^* \in \mathcal{S}_i^{(1)}$ be the vertex such that $\mathsf{s}^\star = \arg\min_{\mathsf{s} \in \mathcal{S}_i^{(1)}} d(\mathsf{c}, \mathsf{s})$.

Define

$$\mathfrak{B} = \{\mathsf{v} \in \mathcal{N}_\mathsf{c} \cap \mathsf{V}_i : \mathrm{POS}_\mathsf{c}(\mathsf{v}, \mathsf{V}_i) < \mathrm{POS}_\mathsf{c}(\mathsf{s}^*, \mathsf{V}_i)\}.$$

Observe that if $\mathsf{v} \in (\mathcal{N}_\mathsf{c} \cap \mathsf{V}_i) \setminus \mathfrak{B}$, then $d(\mathsf{v}, \mathsf{c}) \geq d(\mathsf{s}^*, \mathsf{c})$ and by triangle inequality $d(\mathsf{v}, \mathsf{s}^*) \leq 2d(\mathsf{v}, \mathsf{c})$. Hence, every $\mathsf{v} \in \mathsf{V}_i \setminus \mathfrak{B}$ is good. Therefore, $\mathsf{V}_i^b \cap \mathcal{N}_\mathsf{c} \subseteq \mathfrak{B}$ and $|\mathcal{N}_\mathsf{c} \cap \mathsf{V}_i^b| \leq \mathrm{POS}_c(\mathsf{s}^*, \mathsf{V}_i)$.

For any constant $\gamma \leq c_{\mathsf{S}1}$,

$$\Pr_{\mathcal{S}_i}\left[|\mathsf{V}_i^b \cap \mathcal{N}_\mathsf{c}| \geq \frac{|\mathsf{V}_i|}{k\gamma \log n}\right] \leq \Pr_{\mathcal{S}_i}\left[\mathrm{POS}_c(\mathsf{s}^*, \mathsf{V}_i) \geq \frac{|\mathsf{V}_i|}{k\gamma \log n}\right] \leq \left(1 - \frac{1}{k\gamma \log n}\right)^{|\mathcal{S}_i^{(1)}|} \leq \frac{1}{n^{\Omega(1)}}.$$

Union bounding over $c \in \mathsf{C}^\star$ gives

$$\Pr_{\mathcal{S}_i}\left[|\mathsf{V}_i^b| \geq \frac{|\mathsf{V}_i|}{\gamma \log n}\right] \leq \frac{1}{n^{\Omega(1)}}.$$

$\square$

### B.3.2 PUTTING EVERYTHING TOGETHER

Define the event IDEAL as all the conditions in all preceding lemmas hold. We perform subsequent analysis under this assumption. We prove structural lemmas: in each round, there are sufficiently many surviving good vertices to control the cost of the removed bad ones (Lemma B.9), and across all rounds we can construct an injection from bad to good vertices (Lemma B.10). We use these structural guarantees to bound the total cost by $O(1) \cdot \mathsf{OPT}_k^1(\mathsf{V})$ (Corollary B.11).

**Lemma B.8.** *With probability at least $1 - n^{-\Omega(1)}$, the event IDEAL holds; i.e., all properties from Lemmas B.2,B.3, B.4, B.5,B.6,B.7 holds simultaneously across all $r = O(\log n)$ rounds.*

Let $\mathcal{R}_i = \mathsf{V}_i'' \cup \mathcal{S}_i^{(1)} \cup \mathcal{S}_i^{(2)}$ be the set of vertices removed in round $i$. Define $\bar{\mathsf{V}}_i^b := \mathsf{V}_i^b \cap \mathsf{V}_i''$ to be the subset of bad vertices that actually get included in the safe-set in round $i$, and let $\bar{\mathsf{V}}^b := \bigcup_{i=1}^r \bar{\mathsf{V}}_i^b$ denote the collection of all such vertices across all rounds. We mainly need to worry about these. Similarly, define $\bar{\mathsf{V}}_i^g := \mathsf{V}_i^g \setminus \mathcal{R}_i$ as the set of good vertices that are not included in the safe-set in round $i$, and let $\bar{\mathsf{V}}^g := \bigcup_{i=1}^r \bar{\mathsf{V}}_i^g$ be the union of all such good vertices.

**Lemma B.9.** *Conditioned on the event IDEAL, the following hold for every round $i$:*

1. $|\bar{\mathsf{V}}_i^g| \geq |\mathsf{V}_i|/100.$

2. *For any $\mathsf{b} \in \bar{\mathsf{V}}_i^b$ and $\mathsf{g} \in \bar{\mathsf{V}}_i^g$,*  $d(\mathsf{b}, \mathcal{M}_i(\mathsf{b})) \leq 4 d(\mathsf{g}, \mathsf{C}^\star).$

*Proof.* By Lemma B.4, $|\mathsf{V}_i'| \geq \frac{3}{5}|\mathsf{V}_i|$. Since the safe-set removes exactly $|\mathsf{V}_i|/4$ vertices, it follows that

$$|\mathsf{V}_i' \setminus \mathsf{V}_i''| \geq \tfrac{3}{5}|\mathsf{V}_i| - \tfrac{1}{4}|\mathsf{V}_i| = \tfrac{7}{20}|\mathsf{V}_i|.$$

By Lemma B.7, at most $O(|\mathsf{V}_i|/\log n)$ of these are bad, and since $\mathcal{R}_i = \mathsf{V}_i'' \cup \mathcal{S}_i^{(1)} \cup \mathcal{S}_i^{(2)}$ by ensuring $n_i > \Omega(m_{\mathsf{S}1}), \Omega(m_{\mathsf{S}2})$ and, we can ensure that

$$|\bar{\mathsf{V}}_i^g| = |\mathsf{V}_i^g \setminus \mathcal{R}_i| = |(\mathsf{V}_i' \setminus \mathcal{R}_i) \setminus \mathsf{V}_i^b| \geq \frac{7}{20}|\mathsf{V}_i| - O\left(\frac{|\mathsf{V}_i|}{\log n}\right) \geq |\mathsf{V}_i|/100.$$

always holds for suitably large $n$.

Next, recall that $V_i''$ consists of vertices that appear among the the $\lfloor |V_i|/4 \rfloor$ edges in $\pi_{\mathcal{N}_i}$. Since $\pi_{\mathcal{N}_i}$ is 4-sorted (Lemma 2.2), for any $u \in V_i''$ and any $w \in V_i' \setminus V_i''$ we have

$$d(\mathfrak{f}_u) \leq 4\, d(\mathfrak{f}_w).$$

By construction of $\mathcal{M}_i$, this yields

$$d(u, \mathcal{M}_i(u)) \leq 4\, d(w, C^\star) \qquad \text{for any } u \in V_i'', \ w \in V_i' \setminus V_i''.$$

Since $\bar{V}_i^g \subseteq V_i' \setminus V_i''$ and $\bar{V}_i^b \subseteq V_i'$, this implies that for any $g \in \bar{V}_i^g$ and $b \in \bar{V}_i^b$

$$d(b, \mathcal{M}_i(b)) \leq 4\, d(g, C^\star).$$

Note that in the case of the last round $V_r^b = \emptyset$ and the claim vacuously holds.

$\square$

**Lemma B.10.** *Conditioned on the event* IDEAL, *there exists a map* $\psi : \bar{V}^b \to \bar{V}^g$ *such that:*

    *(i) for every* $b \in \bar{V}_i^b$, *the image satisfies* $\psi(b) \in \bar{V}_i^g$ *(i.e. each removed bad vertex is mapped to a surviving good vertex from the same round),*

    *(ii)* $\psi$ *is an injection.*

*Proof.* By Lemma B.7, in every round $i$, $|V_i^b \cap V_i''| \leq \frac{|V_i|}{100 \log n}$, while by Lemma B.9, $\bar{V}_i^g \subseteq (V_i' \setminus \mathcal{R}_i) \cap V_i^g$ and of size $|\bar{V}_i^g| \geq \frac{|V_i|}{100}$. Moreover, since each round removes exactly a $\frac{1}{4}$ fraction of $V_i$, we have

$$|V_{i+1}| \leq \frac{3}{4}\, |V_i|.$$

We construct $\psi$ by reverse induction over the rounds. For the final round $i = r$, the claim holds by default as there are no bad vertices, i.e. $V_r^b = \emptyset$.

Assume injections $\psi_j$ are defined for all $j > i$, such that the injection property is preserved so far. Let

$$U := \bigcup_{j=i+1}^{r} \psi_j(V_j^b \cap V_j''),$$

denote the set of *used* vertices. Therefore,

$$|U| \leq \sum_{j=i+1}^{r} \frac{|V_j|}{100 \log n} \leq \frac{|V_i|}{100 \log n} \sum_{t=1}^{\infty} (3/4)^t = \frac{3|V_i|}{100 \log n}.$$

Now define $V_i^{\text{rem}} := \bar{V}_i^g \setminus U$. Then

$$|V_i^{\text{rem}}| \geq \frac{|V_i|}{100} - \frac{3|V_i|}{100 \log n} \geq \frac{|V_i|}{150} \geq \frac{|V_i|}{100 \log n} \geq |V_i^b| \geq |\bar{V}_i^b|$$

for sufficiently large $n$. Thus an injection $\psi_i : \bar{V}_i^b \to V_i^{\text{rem}}$ exists, and its image is disjoint from $U$. This completes the proof. $\square$

**Corollary B.11.** *Conditioned on the event* IDEAL, $\sum_{v \in V} d(v, \mathcal{M}(v)) = O(1) \cdot \text{OPT}_k^1(V)$.

*Proof.* By Lemma B.10 and Lemma B.9 there exists an injection $\psi : \bar{V}^b \to \bar{V}^g$ such that $d(b, \mu(b)) \leq 4\, d(\Psi(b), C^\star)$.

Summing over $\bar{V}^b$ and using that $\psi$ is injective,

$$\sum_{b \in \bar{V}^b} d(b, \mathcal{M}(b)) \leq 4 \sum_{b \in \bar{V}^b} d(\Psi(b), C^\star) \leq 4 \sum_{v \in V} d(v, C^\star) = 4\, \text{OPT}_k^1(V).$$

Note that by the definition of "good", for every $v \in V \setminus V_i^b$ must have been good, i.e.,

$$d(v, \mathcal{M}(v)) \le \max\{\mathcal{L}, 4\, d(v, C^\star)\}, \qquad \mathcal{L} := \mathsf{OPT}_k^1(V)/n.$$

Hence

$$\sum_{g \in V \setminus \bar{V}^b} d(v, \mu(v)) \le \sum_{v \in V \setminus \bar{V}^b} \max\{\mathcal{L}, C\, d(g, C^\star)\} \le 4\, \mathsf{OPT}_k^1(V) + n \cdot \frac{\mathsf{OPT}_k^1(V)}{n} = 5\, \mathsf{OPT}_k^1(V).$$

Taken together,

$$\sum_{v \in V} d(v, \mathcal{M}(v)) = O(1) \cdot \mathsf{OPT}_k^1(V),$$

$\square$

**Query Complexity.** Since PROBSORT$(\cdot)$ is always invoked on a set of size $m_{\mathsf{S1}} \cdot m_{\mathsf{S2}} = O(k^2 \text{polylog}\, n)$, each call uses $O(k^2 \text{polylog}\, n)$ queries. Computing proximity scores requires one evaluation per $(s, v)$ pair, i.e. $O(|\mathcal{S}_i^{(1)}| \cdot |V_i|) = O(nk\text{polylog}\, n)$ queries in a round. Each invocation of ALG-TESTER occurs within ADVSORT on $\mathcal{Y}_i = \mathcal{E}(\mathcal{S}_i^{(1)}, V_i')$, contributing another $O(nk\text{polylog}\, n)$ queries. Therefore, the total per-round query complexity is $O((nk + k^2)\text{polylog}\, n) = O(nk \cdot \text{polylog}\, n)$. Since there are $r = O(\log n)$ rounds, the overall query complexity is $O(nk \cdot \text{polylog}\, n)$.

Notice that all results, in this section, up to constant factors in asymptotic analysis, can be extended to any $(k, p)$-clustering instance, where $p$ is a constant or $\infty$. We conclude with Theorem 3.1.

## C   BOUNDED DOUBLING DIMENSION

In this section, we show how to improve the query complexity of ALG-G to $O((n + k^2)\, \text{polylog}\, n)$ when the underlying metric has a bounded doubling dimension. We use the following data structure and query procedure from Raychaudhury et al. (2025).

**Lemma C.1.** *Let $\Sigma = (V, d)$ be a metric of bounded doubling dimension with $|V| = n$, and $\mathcal{E}(V) = \mathcal{E}(V, V)$ accessible under the R-model. For $\mathcal{S} \subseteq V$ and an $\alpha$-sorted ordering $\pi_{\mathcal{E}(\mathcal{S})}$, there exists a procedure CONSTRUCT$(\mathcal{S}, \pi_{\mathcal{E}(\mathcal{S})})$ that builds a structure $\mathcal{T}$ without executing any quadruplet oracle, such that given any $v \in V \setminus \mathcal{S}$, a procedure TRAVERSE$(\mathcal{T}, v)$ returns a set $\mathcal{F} \subseteq \mathcal{E}(v, \mathcal{S})$ of size $O(\text{polylog}\, n)$ such that $\min_{e \in \mathcal{F}} d(e) \le 4\alpha \cdot d(v, \mathcal{S})$. The traversal requires answers to $O(\text{polylog}\, n)$ quadruplet queries of the form $(\{s_1, s_2\}, \{s_3, v\})$ with $s_i \in \mathcal{S}$ from a adversarial quadruplet oracle with noise $\mu \le 1$.*

### C.1   ALG-D

Similar to the general metric case, the algorithm proceeds in rounds. We describe the steps of a generic round $i$. The first three steps are identical to ALG-G, but we restate them for convenience.

1. *Sampling.* Sample uniformly at random a set of vertices $\mathcal{S}_i^{(1)}, \mathcal{S}_i^{(2)} \subseteq V_i$ of sizes $m_{\mathsf{S1}} = c_{\mathsf{S1}} k \log^2 n$ and $m_{\mathsf{S2}} = c_{\mathsf{S2}} k \log^3 n$ respectively, where $c_{\mathsf{S1}}, c_{\mathsf{S2}}$ are suitable constants. Let $\mathcal{S}_i := \mathcal{S}_i^{(1)} \cup \mathcal{S}_i^{(2)}$.

2. *Order edges.* Let $\mathcal{X}_i = \mathcal{E}(\mathcal{S}_i^{(1)}, \mathcal{S}_i^{(2)})$ and compute $\pi_{\mathcal{X}_i} = \text{PROBSORT}(\mathcal{X}_i)$.

3. *Kernel and guard sets.* Let $m_{\mathsf{win}} = 2\max\{c_{\mathsf{win}} \log n, \mathcal{D}\}$. For each $s \in \mathcal{S}_i^{(1)}$, let $\mathcal{X}_{i,s} = \mathcal{E}(s, \mathcal{S}_i^{(2)})$ and $\pi_{\mathcal{X}_{i,s}}$ the ordering of $\mathcal{X}_{i,s}$ induced by $\pi_{\mathcal{X}_i}$. For every $s \in \mathcal{S}_i^{(1)}$, compute

$$\text{KERNEL}_i(s) = \{w \in \mathcal{S}_i^{(2)} : \text{RANK}_{\pi_{\mathcal{X}_{i,s}}}[\{s, w\}] \le m_{\mathsf{win}}\},$$

$$\text{GUARD}_i(s) = \{g \in \mathcal{S}_i^{(2)} : m_{\mathsf{win}} + \mathcal{D} < \text{RANK}_{\pi_{\mathcal{X}_{i,s}}}[\{s, g\}] \le 2m_{\mathsf{win}} + \mathcal{D}\}.$$

4. *Identify close pairs.* For each $s \neq s' \in \mathcal{S}_i^{(1)}$, compute $\text{PCOUNT}_s(s')$. Define $\{s, s'\}$ as *close* if

$$\max\{\text{PCOUNT}_s(s'), \text{PCOUNT}_{s'}(s)\} \geq \lfloor m_{\text{win}}/2 \rfloor.$$

5. *Partition into classes.* Construct a graph $G_i$ on $\mathcal{S}_i^{(1)}$ whose edges are all close pairs as defined above. As we show in the analysis, the graph $G_i$ is $O(\log n)$-*degenerate*, i.e., in any subgraph of $G_i$ there exists at least one vertex with degree $O(\log n)$. We run the greedy coloring algorithm for degenerate graphs Lick & White (1970) on $G_i$ to get a coloring of the vertices in $G_i$ and let $\chi_i$ be the number of different colors. We partition $\mathcal{S}_i^{(1)}$ into classes $\mathcal{S}_i^{(1,1)}, \ldots, \mathcal{S}_i^{(1,\chi_i)}$ based on their colors. Let $\mathcal{E}_i^{(j)} = \{(u, v) \mid u, v \in \mathcal{S}_i^{(1,j)}\}$ for each $j = 1, \ldots, \chi_i$.

6. *Approximate nearest neighbors.* For each class $\mathcal{S}_i^{(1,j)}$:

   (a) *Build.* Compute $\pi_{\mathcal{E}_i^{(j)}} = \text{ADVSORT}(\mathcal{E}_i^{(j)})$ with ALG-TESTER as the comparator, then construct $\mathcal{T}_i^{(j)} = \text{CONSTRUCT}(\mathcal{S}_i^{(1,j)}, \pi_{\mathcal{E}_i^{(j)}})$.

   (b) *Traverse.* For each $v \in V_i \setminus \mathcal{S}_i$, run $\text{TRAVERSE}(\mathcal{T}_i^{(j)}, v)$. During execution, the procedure issues comparisons of the form $(\{s_1, s_2\}, \{s_3, v\})$ with $s_1, s_2, s_3 \in \mathcal{S}_i^{(1,j)}$. For each comparison, first check whether

   $$\max\{\text{PCOUNT}_{s_1}(v), \text{PCOUNT}_{s_2}(v), \text{PCOUNT}_{s_3}(v)\} \geq \lfloor m_{\text{win}}/2 \rfloor.$$

   If the condition holds, eliminate $v$ and proceed to the next vertex. Otherwise, call $\text{ALG-TESTER}(\{s_1, s_2\}, \{s_3, v\})$ and pass its response to TRAVERSE. If $v$ is not eliminated, TRAVERSE outputs $O(\text{polylog } n)$ edges from $\mathcal{E}(v, \mathcal{S}_i^{(1,j)})$.

7. *Collect results.* Let $\mathcal{Y}_i^{(j)}$ be the set of edges returned for class $j$ in the previous step, and define $\widehat{\mathcal{Y}}_i = \bigcup_{j=1}^{\chi_i} \mathcal{Y}_i^{(j)}$. For every eliminated vertex $v$, remove all incident edges from $\widehat{\mathcal{Y}}_i$, and denote the remaining set by $\mathcal{Y}_i$. Let $V_i' := V(\mathcal{Y}_i) \setminus \mathcal{S}_i$.

8. *Final ordering.* Compute $\pi_{\mathcal{Y}_i} = \text{ADVSORT}(\mathcal{Y}_i)$ using ALG-TESTER as the comparator. For each $v \in V_i'$, let $\mathbb{f}_v$ be the first edge incident to $v$ in $\pi_{\mathcal{Y}_i}$.

9. *Safe-set and mapping.* Let $\pi_{\mathcal{N}_i}$ be the ordering of $\mathcal{N}_i$ induced by $\pi_{\mathcal{Y}_i}$. Define $V_i'' = \{v \in V_i' : \text{RANK}_{\pi_{\mathcal{N}_i}}(\mathbb{f}_v) \leq |V_i|/4\}$. For every $v \in V_i''$, define $\mathcal{M}_i(v)$ as the endpoint of $\mathbb{f}_v$ in $\mathcal{S}_i^{(1)}$. For every $v \in \mathcal{S}_i$, define $\mathcal{M}_i(v) := v$.

10. *Recurse.* Set $V_{i+1} = V_i \setminus (V_i'' \cup \mathcal{S}_i)$ and proceed to next round.

## C.2 ANALYSIS

**Lemma C.2.** *In any round $i$, conditioned on Lemma B.2 with probability $1 - n^{-4}$, $\chi_i = O(\log n)$.*

*Proof.* Fix a round $i$. By Lemma B.3, we can argue that w.h.p. for every $\{s, s'\} \in \mathcal{E}(\mathcal{S}_i^{(1)})$, if $\{s, s'\}$ is close then $\text{POS}_s(s', V_i) \leq |V_i|/(100 m_{\text{S1}})$ or $\text{POS}_{s'}(s, V_i) \leq |V_i|/(100 m_{\text{S1}})$.

For any $s \in \mathcal{S}_i^{(1)}$ define $\deg^+(s) := \left|\{s' \in \mathcal{S}_i^{(1)} \setminus \{s\} : \text{POS}_s(s', V_i) \leq |V_i|/(100 m_{\text{S1}})\}\right|$. We note that $\deg^+(s)$ is not the degree of $s$ in $G_i$.

Recall $m_{\text{S1}} = c_{\text{S1}} k \log^2 n$. For a uniformly random $s' \in V_i \setminus \{s\}$,

$$\Pr\left[\text{POS}_s(s'; V_i) \leq \frac{|V_i|}{100 m_{\text{S1}}}\right] \leq \frac{1}{100 m_{\text{S1}}}.$$

Over the $m_1 - 1$ (with-replacement) draws in $\mathcal{S}_i^{(1)} \setminus \{s\}$, $\mathbb{E}[\deg^+(s)] \leq \frac{1}{100}$. By the Chernoff bound, for any constant $c \geq 10$ and large enough $n$, $\Pr[\deg^+(s) \geq C \log n] \leq n^{-3}$. By union bound for every $s \in \mathcal{S}_i^{(1)}$ with probability at least $1 - \frac{1}{n^2}$ it holds that $\deg^+(s) = O(\log n)$.

From the above argument, we can conclude that for every $s \in \mathcal{S}_i^{(1)}$ can contribute to at most $O(\log n)$ edges being close. Thus, with high probability, for any subgraph $H$ of $G_i$, the total number

of edges is bounded by $O(\log n) \cdot |V(H)|$. Thus, for every subgraph $H$ of $G_i$ there is at least some vertex with degree at most $O(\log n)$, and the graph $G_i$ is $O(\log n)$-degenerate Lick & White (1970). It is known that the greedy coloring according to the degeneracy ordering can color graph $G_i$ with $\chi_i = O(\log n)$ colors Lick & White (1970).

$\square$

**Lemma C.3.** *In any round $i$, conditioned on Lemma B.2, with probability $1 - n^{-3}$, whenever* ALG-TESTER *is invoked, it behaves like an adversarial quadruplet oracle with error $\mu = 1$.*

*Proof.* Conditioned on Lemma B.2, each of Lemma B.3 and Lemma B.5 holds with probability $1 - n^{-4}$. By a union bound, they simultaneously hold with probability $1 - n^{-3}$. We perform the analysis under this assumption.

By Lemma B.5, ALG-TESTER behaves like an adversarial oracle with error $\mu = 1$ if for any query $q = (\{s'_1, v'_1\}, \{s'_2, v'_2\})$, where $s'_1, s'_2 \in \mathcal{S}_i^{(1)}$ and $v'_1, v'_2 \in V_i \setminus \mathcal{S}_i^{(2)}$, it holds that

$$d(v'_j, s'_j) > \max_{w \in \text{KERNEL}_i(s_j)} d(s'_j, w) \qquad (j = 1, 2). \tag{2}$$

By Lemma B.3, if $d(s, v) \leq \max_{w \in \text{KERNEL}_i(s)} d(s, w)$ then $\text{PCOUNT}_s(v) > \lfloor m_{\text{win}}/2 \rfloor$.

*Construction phase.* Every call compares $\{s_1, s_2\}, \{s_3, s_4\}$ with $s_\ell \in \mathcal{S}_i^{(1,j)}$. Since each $\mathcal{S}_i^{(1,j)}$ is an independent set of $G_i$ we have that $s_1$ is not close to $s_2$ and $s_3$ is not close to $s_4$. Consider the pair $\{s_1, s_2\}$. Since it is not close, we have $\max\{\text{PCOUNT}_{s_1}(s_2), \text{PCOUNT}_{s_2}(s_1)\} < \lfloor m_{\text{win}}/2 \rfloor$. Without loss of generality, assume that $\text{PCOUNT}_{s_1}(s_2) < \lfloor m_{\text{win}}/2 \rfloor$. From Lemma B.3, we have that $d(s_1, s_2) > \max_{w \in \text{KERNEL}_i(s_1)} d(s_1, w)$. Similarly, the same inequality holds for the pair $\{s_3, s_4\}$, and hence (2) is satisfied.

*Traverse phase.* Queries are of the form $(\{s_1, s_2\}, \{s_3, v\})$ with $s_\ell \in \mathcal{S}_i^{(1,j)}$ and $v \in V_i \setminus \mathcal{S}_i$. Before invoking ALG-TESTER, the algorithm ensures $\text{PCOUNT}_{s_3}(v) \leq \lfloor m_{\text{win}}/2 \rfloor$, which implies $d(v, s_3) > \max_{w \in \text{KERNEL}_i(s_3)} d(s_3, w)$. Combined with the argument for $\{s_1, s_2\}$ above, the tester precondition (2) holds.

*Final sorting.* Queries are of the form $(\{s_1, v_1\}, \{s_2, v_2\})$ with $s_1, s_2 \in \mathcal{S}_i^{(1,j)}$ and $v_1, v_2 \in V'_i$. Since any such $\{s, v\}$ must have passed the traverse step, we have $\text{PCOUNT}_s(v) \leq \lfloor m_{\text{win}}/2 \rfloor$, and therefore (2) holds.

$\square$

**Lemma C.4.** *In any round $i$ of* ALG-D, *conditioned on Lemma B.2, with probability at least $1 - n^{-3}$, the following holds:*

*(i) For every $v \in V''_i$, $d(v, \mathcal{M}_i(v)) \leq 64\, d(v, \mathcal{S}_i^{(1)})$.*

*(ii) $|V''_i| \geq |V_i|/4$.*

*Proof.* Conditioned on Lemma B.2, each of Lemma B.3, Lemma B.4, and Lemma C.3 holds with probability $1 - n^{-4}$. By a union bound, they simultaneously hold with probability $1 - n^{-3}$. We perform the analysis under this assumption.

By Lemma C.3, all calls to ALG-TESTER are correct. Hence, by Lemma C.1 both the construction steps succeed and every traverse step (for non-eliminated vertices) is successful for every vertex in $\mathcal{S}_i^{(1)}$. From Lemmas C.3 and 2.2, the ordering $\pi_{\mathcal{E}_i^{(j)}}$ is 4-sorted, for every $j = 1, \ldots, \chi_i$. From Lemma C.1, for every vertex $v \in V_i \setminus \mathcal{S}_i$, the set $\mathcal{Y}_i$ contains an edge $e_{v,j} \in \mathcal{E}(v, \mathcal{S}_i^{(1,j)})$, for each class $\mathcal{S}_i^{(1,j)}$, such that $d(e_{v,j}) \leq 16 d(v, \mathcal{S}_i^{(j)})$. Similarly, from Lemmas C.3 and 2.2 the ordering $\pi_{\mathcal{Y}_i}$ is 4-sorted.

Let $\mathfrak{f}_v$ be the lowest rank edge incident to $v \in V'_i$ in $\pi_{\mathcal{Y}_i}$, and recall $\mathcal{M}_i(v)$ is its endpoint in $\mathcal{S}_i^{(1)}$. Then for every $v \in V''_i$,

$$d(v, \mathcal{M}_i(v)) = d(\mathfrak{f}_v) \leq 4 \min_{e_{v,j} \in \mathcal{Y}_i} d(e_{v,j}) \leq 64\, d(v, \mathcal{S}_i^{(1)}).$$

For part(ii), observe that any vertex eliminated at any stage must have satisfied $\text{PCOUNT}_s(v) > \lfloor m_{\text{win}}/2 \rfloor$ for some s, which by Lemma B.3 implies $\text{POS}_s(v, V_i) \leq |V_i|/(100m_{\text{S1}})$. Hence every eliminated vertex lies within the lowest-ranked $|V_i|/(100m_{\text{S1}})$ for some s, so in total at most $|V_i|/100$ vertices can be eliminated. Thus, $|V_i'| \geq (3/5)|V_i|$, which immediately implies that $|V_i''| \geq \frac{|V|}{4}$. $\qquad\square$

Using the results of Lemmas C.3, C.4 along with the analysis in Section B.3.2 for general metrics, we conclude that $\sum_{v \in V} d(v, \mathcal{M}(v)) = O(1) \cdot \text{OPT}_k^1(V)$, with high probability.

**Query Complexity.** In step 2 of the algorithm, $\text{PROBSORT}(\mathcal{X}_i)$ uses $O(\max\{k^2, n\}\text{polylog}\, n)$ quadruplet queries since $|\mathcal{X}_i| = O(k^2\text{polylog}\, n)$. For every $s \in \mathcal{S}_i^{(1)}$, $|\text{KERNEL}_i(s)|, \text{GUARD}(s) = O(\text{polylog}\, n)$. Hence, in step 4 the algorithm calls $O(k^2\text{polylog}\, n)$ queries to the quadruplet oracle to compute $\text{PCOUNT}_s(s')$ for every pair $s \neq s' \in \mathcal{S}_i^{(1)} \times \mathcal{S}_i^{(1)}$. In step 6(a), $\sum_{j=1,\ldots,\chi_i} |E_i^{(j)}| = O(k^2\text{polylog}\, n)$ so $\text{ADVSORT}(\mathcal{E}_i^{(j)})$ with $\text{ALG-TESTER}$ as the comparator use $O(k^2\text{polylog}\, n)$ queries to the quadruplet oracle over all classes $\mathcal{S}_i^{(1,j)}$. In step 6(b), for each $v \in V_i \setminus \mathcal{S}_i$, the $\text{TRAVERSE}(\mathcal{T}_i^{(j)}, v)$ (including the computations of $\text{PCOUNT}_{s_h}(v)$) use $O(\text{polylog}\, n)$ queries, so in total step 6(b) executes $O(n\text{polylog}\, n)$ queries to the quadruplet oracle. In step 8, $|\mathcal{Y}_i| = O(n\text{polylog}\, n)$, so the procedure $\text{ADVSORT}(\mathcal{Y}_i)$ using $\text{ALG-TESTER}$ as the comparator, runs $O(n\text{polylog}\, n)$ queries to the quadruplet oracle. All other steps of the algorithm do not execute any quadruplet oracle query. Overall, our algorithm calls the quadruplet oracle $O((n + k^2)\text{polylog}\, n)$ times. We conclude with Theorem 3.2.

# D IMPROVING THE APPROXIMATION QUALITY

We now present a technique for improving the approximation ratio when the underlying metric $\Sigma$ has bounded doubling dimension, $\zeta = O(1)$, while still ensuring that the number of centers used is $\Theta(k \, \text{polylog}\, n)$. In fact, our new procedure also constructs a $(k, \varepsilon)$-coreset. We assume that $\varepsilon \in (0, 1)$ is an arbitrarily small constant.

## D.1 ALG-DI

**Context.** Suppose we have run $\text{ALG-D}(V)$ and obtained $(C, \mathcal{M})$. Let $\bar{r}$ be the total number of rounds in $\text{ALG-D}(V)$ and set $\mathcal{S}^{(1)} := \bigcup_{i=1}^{\bar{r}-1} \mathcal{S}_i^{(1)}$ and $V' := V \setminus C$. By construction, $\mathcal{M}(v) \in \mathcal{S}^{(1)}$ for every $v \in V'$ and $\mathcal{M}(v) = v$ for every $v \in C$.

**Algorithm.** The algorithm proceeds as follows:

1. *Initialization.* Set $\mathcal{Z} \leftarrow \emptyset$.

2. *Per-center processing.* For each $s \in \mathcal{S}^{(1)}$:

   (a) *Assigned set.* Define
   $$U_s := \{ v \in V' \mid \mathcal{M}(v) = s \}.$$
   If s was chosen in round $i$ of $\text{ALG-D}(V)$, then every $v \in U_s$ satisfies $\{s, v\} \in \mathcal{N}_i$. Let $\pi_{U_s}$ be the order on $U_s$ induced by $\pi_{\mathcal{N}_i}$.

   (b) *Level sets.* For $t = 0, 1, 2, \ldots, \lceil \log |U_s| \rceil$ define
   $$U_s^t := \{ v \in U_s \mid \text{RANK}_{\pi_{U_s}}(v) \geq |U_s|/2^t \}.$$
   Let
   $$t_s := \min\{ t \geq 1 \mid |U_s^t| \leq c_{\text{IMP}} \log^3 n \},$$
   where $c_{\text{IMP}}$ is a sufficiently large constant that depends on $\varepsilon$ and $\zeta$.

   (c) *Sampling.* For every $t < t_s$, sample with replacement a subset $W_s^t \subseteq U_s^t$ of size $|W_s^t| = c_{\text{IMP}} \log^3 n$. Set
   $$W_s := \left( \bigcup_{t=0}^{t_s-1} W_s^t \right) \cup U_s^{t_s}.$$

(d) *Edge sets.* For every $t < t_s$, compute

$$\mathcal{Z}_s^t := \mathcal{E}\left(U_s^t \setminus W_s, W_s\right),$$

and update

$$\mathcal{Z} \leftarrow \mathcal{Z} \cup \mathcal{Z}_s^t.$$

3. *Augment centers.* Let $W = \bigcup_{s \in \mathcal{S}^{(1)}} W_s$. Set $C^+ := C \cup W$. For every $v \in C^+$, set $\mathcal{M}^+(v) := v$.

4. *Ordering.* Compute $\pi_{\mathcal{Z}} := \text{PROBSORT}(\mathcal{Z})$.

5. *Final mapping.* For each $v \in V'$:

   (a) Let $s := \mathcal{M}(v)$ (the old mapping). Define $\mathcal{E}_v := \mathcal{E}(v, W_s)$. Let $\pi_{\mathcal{E}_v}$ be the ordering of $\mathcal{E}_v$ induced by $\pi_{\mathcal{Z}}$.

   (b) Let $\{v, w^\star\}$ be the first edge of $\pi_{\mathcal{E}_v}$; note that $w^\star \in W_s$. Set $\mathcal{M}^+(v) := w^\star$.

## D.2 ANALYSIS OF ALG-DI

First, it is straightforward that ALG-DI satisfies the isolation property, i.e., no quadruplet query is ever repeated between ALG-D ALG-DI. We note that all quadruplet oracles executed in ALG-D involved at least one vertex from C. However, in ALG-DI, after the execution of ALG-D, the set C is removed from $V'$, so no quadruplet query will be repeated.

**Query Complexity.** From the analysis of ALG-D, we showed that we executed $O((n + k^2)\text{polylog } n)$ queries to the quadruplet oracle. Then, in step (4) of ALG-DI we execute PROBSORT($\mathcal{Z}$). The set $\mathcal{Z}$ contains $O(k\text{polylog } n)$ edges, so we execute $O(n\text{polylog } n)$ additional queries to the quadruplet oracle. Overall, ALG-DI executes $O((n + k^2)\text{polylog } n)$ queries to the quadruplet oracle.

### D.2.1 FOCUSING ON A FIXED VERTEX

In the following, we focus on a fixed $s \in \mathcal{S}^{(1)}$. Let $|U_s| = m$. Suppose $\sum_{u \in U_s} d(s, u) = m\mathcal{L}$ for a suitable $\mathcal{L} \geq 0$. This also implies that $\max_{u \in U_s} d(s, u) \leq m\mathcal{L}$.

**Fixed buckets.** We know from the analysis in Section C.2 that $\pi_{U_s}$ is 4-sorted. Based on the order $\pi_{U_s}$, define a contiguous partition of $U_s$ into buckets $\{B_t\}_{t=0}^b$, where $b$ is defined next, as follows. Let $\delta = \frac{\varepsilon}{100 c_{app}}$, where $c_{app}$ is the approximation ratio of ALG-D. For $t = 0, 1, 2, \ldots$, let $B_t$ be the next consecutive block in the order ending at the rightmost element whose distance from s is at most $2^t \delta \mathcal{L}$:

$$B_0 = \{\text{initial consecutive block up to the last u with } d(s, u) \leq \delta\mathcal{L}\},$$

$$B_1 = \{\text{next block up to the last u with } d(s, u) \leq 2\delta\mathcal{L}\}, \quad \ldots$$

Since every $u \in U_s$ satisfies $d(s, u) \leq m\mathcal{L}$, the number of buckets is

$$b = O\left(\log \frac{m\mathcal{L}}{\delta\mathcal{L}}\right) = O(\log \frac{m}{\delta}) = O(\log \frac{n}{\delta}) = O(\log n).$$

For the sake of analysis, consider the following recursive partitioning of $\{B_t\}_{t=1}^b$ (note that we exclude $B_0$):

**Partitioning.** Let $i_0 := 0$. For rounds $r = 1, 2, \ldots$ do:

1. Compute $m_r := \sum_{t=i_{r-1}+1}^b |B_t|$. If $m_r \leq c_{\text{IMP}} \log^3 n$, then stop, else proceed.

2. Define the *heavy threshold* at round $r$ by $\tau_r := \frac{m_r}{100 \log n}$

3. Let $i_r$ be the largest index $t \in \{i_{r-1}+1, \ldots, b\}$ such that the bucket $B_t$ contains at least $\tau_r$ vertices.

4. Remove the entire prefix of fixed buckets $\{B_t\}_{t=i_{r-1}+1}^{i_r}$

We now prove certain properties of the above partitioning. Let $r^\star$ be the index of the last round.

**Lemma D.1.** *In every round $r$, if $m_r \geq c_{\mathsf{IMP}} \log^3 n$, there exists an index $i_r \in \{i_{r-1} + 1, \ldots, b\}$ such that $|B_{i_r}| \geq \tau_r = \frac{m_r}{100 \log n}$. Furthermore, $m_{r+1} \leq \frac{m_r}{100}$ and $r^\star \leq \log n$.*

*Proof.* Let $b_r := b - i_{r-1}$ be the number of active buckets in round $r$. If every active bucket was strictly smaller than $\frac{m_r}{100 \log n}$, then

$$m_r = \sum_{t=i_{r-1}+1}^{b} |B_t| < b_r \cdot \frac{m_r}{100 \log n} \leq \log n \cdot \frac{m_r}{100 \log n} = \frac{m_r}{100},$$

which is a contradiction. Hence, a heavy bucket exists; let $i_r$ be the largest index with $|B_{i_r}| \geq \frac{m_r}{100 \log n}$. The above argument also implies that all buckets to the right of $i_r$ can have a total of at most $\frac{m_r}{100}$ vertices. Thus, $m_{r+1} \leq \frac{m_r}{100}$ which directly implies that $r^\star \leq \log n$. $\square$

For any round $r < r^\star$, mark an active bucket in round $r$ as *light* if its cardinality is less than $\delta m_r / (1000 \log^2 n)$. Let $\mathrm{LIGHT}_r := \{ t \in \{i_{r-1} + 1, \ldots, i_r - 1\} \mid |B_t| < \delta m_r / (1000 \log^2 n) \}$ be the index set of light buckets in round $r$.

**Lemma D.2.** *For any round $r$, $\sum_{t \in \mathrm{LIGHT}_r} \sum_{\mathsf{v} \in B_t} d(\mathsf{s}, \mathsf{v}) \leq \delta \cdot \sum_{\mathsf{v} \in B_{i_r}} d(\mathsf{s}, \mathsf{v})$.*

*Proof.* Since $\pi_{\mathsf{U_s}}$ is 4-sorted and the buckets $\{B_t\}$ are contiguous in the order, for any $t \in \mathrm{LIGHT}_r$, for every $\mathsf{v}' \in B_t$, and any $\mathsf{v} \in B_{i_r}$, $d(\mathsf{s}, \mathsf{v}') \leq 4 d(\mathsf{s}, \mathsf{v})$.

Let $\bar{\mathsf{v}} = \arg\min_{\mathsf{v} \in B_{i_r}} d(\mathsf{v}, \mathsf{s})$. Therefore,

$$\sum_{t \in \mathrm{LIGHT}_r} \sum_{\mathsf{u} \in B_t} d(\mathsf{s}, \mathsf{u}) \leq \left( \sum_{t \in \mathrm{LIGHT}_r} |B_t| \right) \cdot 4d(\bar{\mathsf{v}}, \mathsf{s}) \leq d(\bar{\mathsf{v}}, \mathsf{s}) \cdot \frac{b \delta m_r}{250 \log^2 n} \leq d(\bar{\mathsf{v}}, \mathsf{s}) \cdot \frac{\delta m_r}{250 \log n}.$$

Furthermore, $d(\bar{\mathsf{v}}, \mathsf{s}) \cdot \frac{m_r}{100 \log n} \leq \sum_{\mathsf{v} \in B_{i_r}} d(\mathsf{s}, \mathsf{v})$, so the result follows.

$\square$

**Lemma D.3.** *With probability at least $1 - n^{-10}$, for every non-light bucket $B$ (i.e., $|B| \geq \delta m_r / (1000 \log^2 n)$ in some round $r$, ALG-DI uniformly samples at least $\mathrm{SIZE} = \gamma \log n$ vertices from $B$, where $\gamma$ is a suitable constant.*

*Proof.* Consider a *non-light* bucket $B$ in some round $r$. Thus, it has at least $\frac{\delta m_r}{1000 \log^2 n}$ vertices. Consider the level sets used by ALG-D with respect to $\mathsf{s}$,

$$\mathsf{U_s}^t = \{ u \in \mathsf{U_s} : \mathrm{RANK}_{\pi_{\mathsf{U_s}}}(u) \geq |\mathsf{U_s}|/2^t \}.$$

Choose $t_B = \max\{ t : B \subseteq \mathsf{U_s}^t \}$. By maximality of $t_B$ and the contiguous nature of the buckets, $|\mathsf{U_s}^{t_B}| \leq 2m_r$. For level $t_B$ the ALG-DI samples uniformly (with replacement) a set $\mathsf{W_s}^{t_B}$ of size $c_{\mathsf{IMP}} \log^3 n$ uniformly from $\mathsf{U_s}^{t_B}$. Let $X$ be the number of sampled vertices that fall in $B$. Then

$$\mathbb{E}[X] = c_{\mathsf{IMP}} \log^3 n \cdot \frac{|B|}{|\mathsf{U_s}^{t_B}|} \geq c_{\mathsf{IMP}} \log^3 n \cdot \frac{\frac{\delta m_r}{1000 \log^2 n}}{2m_r} = \frac{\delta c_{\mathsf{IMP}}}{2000} \log n.$$

By a Chernoff bound, for sufficiently large $c_{\mathsf{IMP}}$, $\Pr[X < \mathrm{SIZE} = \gamma \log n] \leq n^{-12}$. Taking a union bound over all $O(\log^2 n)$ possible non-light buckets across $r^\star = O(\log n)$ rounds proves the claim. $\square$

**Lemma D.4.** *Conditioned on the correctness of* $\mathrm{PROBSORT}(\mathcal{Z})$ *(so that all induced orders used below have maximum dislocation $\mathcal{D}$), there is a choice of absolute constants such that, with probability at least $1 - n^{-8}$, for every non-light bucket $B$ in any round $r$,*

$$\sum_{\mathsf{v} \in B} d(\mathsf{v}, \mathcal{M}^+(\mathsf{v})) \leq \delta \cdot \sum_{\mathsf{v} \in B} d(\mathsf{s}, \mathsf{v}).$$

*Proof.* Fix a non-light bucket $B$ in some round $r$. By the bucket construction and the 4-sorted property of $\pi_{U_s}$, there exists a scale $\alpha > 0$ such that

$$\alpha \;\leq\; d(\mathsf{s}, \mathsf{v}) \;\leq\; 4\alpha \qquad \text{for every } \mathsf{v} \in B.$$

Since the doubling dimension is $\zeta$, it is known that $B$ can be covered by a set of $\widehat{m} = \Theta(\delta^{-\zeta})$ balls of diameter at most $(\delta\alpha)/10$ whose centers lie in $B$. Consider a collection of such balls and partition $B$ in $\widehat{m}$ *components*, breaking ties arbitrarily if two points lie in the same ball. Call a component *light* if it contains fewer than $|B|/\widehat{m}^2$ vertices, and *heavy* otherwise. Then the union of light components contains at most $(1/\widehat{m})\,|B|$ vertices.

*Contribution of heavy components.* Fix a heavy component $B' \subseteq B$. By Lemma D.3 (applied to the present round $r$ and bucket $B$), ALG-DI draws at least SIZE $= \gamma \log n$ independent uniform samples from $B$ (with replacement), where $\gamma > 0$ is a sufficiently large absolute constant. Let $X$ be the number of samples that land in $B'$. Since it contains at least $|B|/\widehat{m}^2$ vertices,

$$\mathbb{E}[X] \;\geq\; \frac{\text{SIZE}}{\widehat{m}^2} \;=\; \frac{\gamma \log n}{\widehat{m}^2}.$$

Assume $\gamma$ is large enough (it depends on $\varepsilon, \zeta$) so that $\mathbb{E}[X] \geq 100 \max\{\log n, \mathcal{D}\}$. Then by a Chernoff bound, $\Pr[X \leq \mathcal{D}] \leq n^{-12}$. By a union bound over all (at most $\widehat{m}$) heavy components, with probability at least $1 - n^{-11}$, each heavy component contains at least $\mathcal{D}$ samples.

Now fix a $\mathsf{v}$ inside $B'$. Recall that $W_\mathsf{s} \subseteq U_\mathsf{s}$ denotes the collection of all vertices including samples ALG-DI computes while processing $\mathsf{s}$. Note that $W_\mathsf{s} \subseteq C^+$. Since all vertices in $B'$ are within distance $\delta\alpha/10$, for every vertex $\mathsf{w} \in W_\mathsf{s} \cap B'$ we have

$$d(\mathsf{v}, \mathsf{w}) \;\leq\; \frac{\delta\alpha}{10}.$$

By the above arguments, $|W_\mathsf{s} \cap B| > \mathcal{D}$.

Since the correctness of PROBSORT($\mathcal{Z}$) implies that the induced order $\pi_{\mathcal{E}_\mathsf{v}}$ (the restriction of $\pi_{\mathcal{Z}}$ to $\mathcal{E}(\mathsf{v}, W)$) has maximum dislocation $\mathcal{D}$, the first edge $\{\mathsf{v}, \mathsf{w}^\star\}$ in $\pi_{\mathcal{E}_\mathsf{v}}$, where $\mathsf{w}^\star \in W \supseteq W_\mathsf{s}$ must satisfy

$$d(\mathsf{v}, \mathsf{w}^\star) = d(\mathsf{v}, \mathcal{M}^+(\mathsf{v})) \leq \frac{\delta\alpha}{10}$$

Summing over all vertices in all heavy components of $B$ contributes at most $(\delta/10) \cdot \alpha \cdot |B|$.

*Contribution of light components.* The union of light contains at most $(1/\widehat{m})\,|B|$ vertices. For any $\mathsf{v}$ in a light component and any sampled $\mathsf{w} \in W_\mathsf{s} \cap B$, the triangle inequality and the bucket scale imply

$$d(\mathsf{v}, \mathsf{w}) \;\leq\; d(\mathsf{v}, \mathsf{s}) + d(\mathsf{s}, \mathsf{w}) \;\leq\; 4\alpha + 4\alpha \;=\; 8\alpha.$$

Since Lemma D.3 guarantees $|W \cap B| \geq$ SIZE $= \gamma \log n \geq \mathcal{D}$, the first edge in $\pi_{\mathcal{E}_\mathsf{v}}$ has length at most $8\alpha$. Therefore, vertices in light components within $B$ contribute at most $(1/\widehat{m}) \cdot |B| \cdot 8\alpha$.

*Putting everything together.* Adding contributions of light and heavy components of $B$, we get that

$$\sum_{\mathsf{v} \in B} d\big(\mathsf{v}, \mathcal{M}^+(\mathsf{v})\big) \;\leq\; \left( \frac{\delta}{10} + \frac{8}{\widehat{m}} \right) |B|\, \alpha.$$

Since $\delta < 1$, $\zeta \geq 1$ and $\widehat{m} = \Theta(\delta^{-\zeta})$, we have that $\frac{8}{\widehat{m}} \leq \frac{\delta}{10}$, and the result follows. The probability bound $1 - n^{-8}$ follows from a union bound over all $O(\log^2 n)$ non-light buckets of $\mathsf{s}$. $\qquad\square$

**Lemma D.5.** *Fix $\mathsf{s} \in \mathcal{S}^{(1)}$. Conditioned on the correctness of* PROBSORT($\mathcal{Z}$), *with probability at least $1 - n^{-8}$,*

$$\sum_{\mathsf{v} \in U_\mathsf{s}} d\big(\mathsf{v}, \mathcal{M}^+(\mathsf{v})\big) \;\leq\; \frac{\varepsilon}{c_{\mathsf{app}}} \cdot \sum_{\mathsf{v} \in U_\mathsf{s}} d(\mathsf{s}, \mathsf{v})\,.$$

*Proof.* We argue for a *single* round $r$ and then sum over rounds. Let $i_r$ be the index of the rightmost heavy bucket in that round. Henceforth we assume Lemma D.2 and Lemma D.4 holds.

By Lemma D.2,

$$\sum_{t \in \text{LIGHT}_r} \sum_{v \in B_t} d(s, v) \ \le \ \delta \cdot \sum_{v \in B_{i_r}} d(s, v).$$

By Lemma D.4 gives, w.h.p., for each *non-light* bucket $B$ in every round,

$$\sum_{v \in B} d(v, \mathcal{M}^+(v)) \ \le \ \delta \cdot \sum_{v \in B} d(s, v).$$

Adding light and non-light contributions within round $r$,

$$\sum_{t=i_{r-1}+1}^{i_r} \sum_{v \in B_t} d(v, \mathcal{M}^+(v)) \le 2\delta \cdot \sum_{v \in B_{i_r}} d(s, v).$$

Since the buckets are disjoint across rounds, summing the above inequality over all rounds, yields

$$\sum_{v \in U_s \setminus B_0} d(v, \mathcal{M}^+(v)) \ \le \ 2\delta \cdot \sum_{v \in U_s \setminus B_0} d(s, v) \ \le \ 2\delta \cdot \sum_{v \in U_s} d(s, v).$$

By design for the bucket $B_0$, $\sum_{v \in B_0} d(v, \mathcal{M}^+(v)) = \sum_{v \in B_0} d(v, s) \le \delta \cdot \sum_{v \in U_s} d(v, s)$. Combining the above, we get that,

$$\sum_{v \in U_s} d(v, \mathcal{M}^+(v)) \ \le \ 3\delta \cdot \sum_{v \in U_s} d(s, v).$$

Since $\delta \ = \ \frac{1}{100} \cdot \frac{\varepsilon}{c_{\text{app}}}$, the stated bound holds. $\qquad\square$

### D.2.2 PUTTING EVERYTHING TOGETHER

**Corollary D.6.** *Conditioned on the correctness of* PROBSORT$(\mathcal{Z})$*, with probability at least* $1 - n^{-7}$ *the refined mapping* $\mathcal{M}^+$ *satisfies*

$$\sum_{v \in V} d(v, \mathcal{M}^+(v)) \ \le \ \varepsilon \cdot \text{OPT}_k^1(V).$$

*Proof.* Fix the event under which Lemma D.5 holds for *every* $s \in \mathcal{S}^{(1)}$. Since each instance holds with probability at least $1 - n^{-8}$ and $|\mathcal{S}^{(1)}| \le n$, a union bound gives overall success probability at least $1 - n^{-7}$. Summing Lemma D.5 over all $s \in \mathcal{S}^{(1)}$ yields

$$\sum_{v \in V \setminus C} d(v, \mathcal{M}^+(v)) \ = \ \sum_{s \in \mathcal{S}^{(1)}} \sum_{v \in U_s} d(v, \mathcal{M}^+(v)) \ \le \ \frac{\varepsilon}{c_{\text{app}}} \cdot \sum_{s \in \mathcal{S}^{(1)}} \sum_{v \in U_s} d(s, v) \ = \ \frac{\varepsilon}{c_{\text{app}}} \cdot \sum_{v \in V \setminus C} d(v, \mathcal{M}(v)).$$

We also know that

$$\sum_{v \in C} d(v, \mathcal{M}^+(v)) \ = \sum_{v \in C} d(v, \mathcal{M}(v)) = 0$$

Since $\mathcal{M}$ is a $c_{\text{app}}$–approximation mapping (from ALG-D), we have

$$\sum_{v \in V} d(v, \mathcal{M}(v)) \ \le \ c_{\text{app}} \cdot \text{OPT}_k^1(V).$$

Putting everything together gives

$$\sum_{v \in V} d(v, \mathcal{M}^+(v)) \ \le \ \frac{\varepsilon}{c_{\text{app}}} \cdot \sum_{v \in V} d(v, \mathcal{M}(v)) \ \le \ \varepsilon \cdot \text{OPT}_k^1(V),$$

as claimed. $\qquad\square$

We conclude with Theorem 3.3.

# E    ADDITIONAL RELATED WORK

Clustering is a classical problem that has been studied for decades Lloyd (1982); Charikar et al. (1999); Har-Peled & Mazumdar (2004); Chen (2009). For $k$-median and $k$-means, Charikar et al. (1999) presented the first constant-factor polynomial-time approximation, and subsequent work Mettu & Plaxton (2002) improved the results. The notion of *coresets* (compact summaries for scalable clustering) for $k$-median and $k$-means clustering was introduced in Har-Peled & Mazumdar (2004) and later advanced by Chen (2009). More recently, results of Braverman et al. (2021; 2022) have improved the sizes of the obtained coresets for clustering.

In another line of work, Xu et al. (2024) studied a weak-strong model in doubling metrics where the weak oracle returns values within a multiplicative factor $C > 1$ of the true distance, while the strong oracle provides exact distances. Their focus, however, was on building data structures for approximate nearest neighbor search. Finally, Silwal et al. (2023) examined correlation clustering within a weak-strong framework, but their techniques and oracle definitions do not extend to the $k$-center/median/means setting.

Much of the oracle-based clustering literature has focused on (faulty or exact) cluster queries Mazumdar & Saha (2017a); Huleihel et al. (2019); Mazumdar & Saha (2017b); Choudhury et al. (2019); Green Larsen et al. (2020); Galhotra et al. (2021), which directly identify ground-truth clusters. For $k$-means Ashtiani et al. (2016); Chien et al. (2018); Kim & Ghosh (2017a;b); Bianchi & Penna (2021), and $k$-median Ailon et al. (2018), such queries, often combined with distance information, have led to stronger approximation guarantees. This is also is closely related to learning-augmented algorithms Fu et al. (2025); Dong et al. (2025); Braverman et al. (2025b; 2024); Grigorescu et al. (2022); Ergun et al. (2022); Mitzenmacher & Vassilvitskii (2022); Indyk et al. (2019); Hsu et al. (2019).

Beyond clustering, distance-based comparison oracles have been applied to a wide range of problems. Examples include learning fairness metrics Ilvento (2020), hierarchical clustering Emamjomeh-Zadeh & Kempe (2018); Chatziafratis et al. (2018); Ghoshdastidar et al. (2019), correlation clustering Ukkonen (2017), classification Tamuz et al. (2011); Hopkins et al. (2020), and knowledge and data engineering Beretta et al. (2023). They have also been leveraged in tasks such as finding the maximum element Guo et al. (2012); Venetis et al. (2012), top-$k$ selection Klein et al. (2011); Polychronopoulos et al. (2013); Ciceri et al. (2015); Davidson et al. (2014); Kou et al. (2017); Dushkin & Milo (2018), information retrieval Kazemi et al. (2018), and skyline computation Verdugo.

Finally, in relational clustering, the objective is to cluster the output of a join (or conjunctive) query. Since the number of join results can be extremely large, materializing them explicitly often leads to prohibitively slow algorithms. To address this, recent works Agarwal et al. (2024); Esmailpour & Sintos (2024); Chen et al. (2022); Moseley et al. (2021); Curtin et al. (2020); Surianarayanan et al. (2025) introduce relational oracles that provide summary statistics of the join output together with access to selected key tuples, enabling clustering to be performed more efficiently. This framework, however, differs from the R-model, where the main challenge lies in evaluating distances between items through noisy comparisons. In relational clustering, by contrast, the difficulty arises primarily from the combinatorial explosion of join results rather than from the complexity of distance evaluation.

## USE OF LARGE LANGUAGE MODELS

Large Language Models (LLMs) were used as general-purpose assist tools. Specifically, we drafted all text ourselves, and in several places (mainly in the introduction) we provided our own written paragraphs to ChatGPT-5 Plus with the instruction to "polish the text." In addition, we used ChatGPT-5 Plus in a limited way to help translate some of our pseudocode into Python code for basic prototyping. No part of the research process, including problem formulation, algorithm design, theoretical development, or experimental analysis, relied on LLMs. All ideas, results, and scientific contributions are entirely our own.

