# OpenReview forum: "Metric $k$-clustering using only Weak Comparison Oracles"
_ICLR.cc/2026/Conference — ICLR 2026 Poster_

### Official Review · Reviewer_yJGX · 2025-10-22

**Soundness:** 3
**Presentation:** 3
**Contribution:** 3
**Rating:** 6
**Confidence:** 3

**Summary:**

The paper studies approximation algorithms for the $(k,p)$-clustering problem in the Rank-model (R-model), where instead of having direct access to pairwise distances, we have access to a (weak) oracle. This oracle takes two pairs of points, $(a,b)$ and $(c,d)$, as input and returns the pair that is closer. Furthermore, the oracle behaves probabilistically: it returns the correct answer with high probability (e.g., 99%) and is persistent. The task is to find a good solution to the $(k,p)$-clustering problem using as few oracle queries as possible.

The paper presents a bicriteria algorithm for the problem that opens $O(k (\log n)^{O(1)})$ centers with an $O(1)$ approximation guarantee, using $O(nk (\log n)^{O(1)})$ queries to the oracle in general metrics. Furthermore, the authors present improved results for metrics of bounded doubling dimension. To achieve their results, they design $O(1)$-“coresets” of size $O(k (\log n)^{O(1)})$, also constructed with $O(nk (\log n)^{O(1)})$ queries, which yields a bicriteria solution. The main technique appears to build on Har-Peled and Mazumdar (STOC ’04), which uses a recursive sampling subroutine to construct such coresets.

**Strengths:**

1. It is known that no constant-factor approximation is possible for $k$-median based solely on the R-model. In this context, the paper presents a promising approach that still achieves a constant-factor approximation, albeit by opening more than $k$ centers.

2. Most of the results appear to extend to the adversarial oracle model, where the oracle is no longer probabilistic but adversarial, giving correct responses only when $d(a,b)$ and $d(c,d)$ are not very close to each other.

3. The presentation is clear and well-organized, especially the technical section that highlights the main ideas of the algorithms.

4. The experimental results are interesting and support the practical relevance of the proposed methods.

**Weaknesses:**

1. Although the paper achieves a constant-factor approximation using $O(k (\log n)^{O(1)})$ centers, it is unclear whether the $\log n$ term is necessary. Ideally, it would be preferable to obtain $O(k)$ centers, or alternatively, provide a lower bound showing that this is not possible.
2. It is not clear how technically different their approach is compared to Har-Peled and Mazumdar, apart from the adaptation to the R-model.
3. It is unfortunate that not a single proof appears in the main paper.

**Questions:**

1. This relates to Weakness #2. I am curious to understand how different the techniques are compared to existing work, particularly what new methods were required to adapt them to the R-model.
2. It would be helpful to include a table summarizing the current state-of-the-art results for the problem (both in the R-model and RM-model) and to clearly highlight the contributions of this paper.
3. On line 67: what is meant by “sublinear approximation”? Could you clarify this term?

---

> ### Author Response · Authors · 2025-11-22
>
> We thank the reviewer for the thoughtful comments.
>
> W1. Great observation! In fact, even in the absence of noise (i.e., the probabilistic error is $0$) we have a proof showing that at least $2k - 1$ centers are necessary to obtain any $o(n)$-approximation algorithm for $k$-median/means clustering using only the weak quadruplet oracle. We included this proof in the appendix of the revised submission.
> In the noisy setting, we believe this lower bound extends to $\omega(k)$ centers. While we do not have a formal lower bound we have some preliminary insights. Suppose we have $k$ tight clusters that are mutually far apart. Suppose we have a set $S$ of $k$ points, one from each cluster. For a new point, we want to determine where it gets mapped. Based on known lower bounds for noisy sorting, we know that using the quadruplet oracle (under probabilistic noise) we cannot even find the nearest neighbor in $S$ within $o(\log n)$ rank. Intuitively, if we want to ensure that the new point gets mapped to the correct cluster, we need to include $\Omega(\log n)$ representatives from each cluster in the set $S$. Intuitively, this suggests that $\Omega(k \log n)$ centers may be necessary. We leave the task of finding a formal lower bound for future work.
>
> W2, Q1. At a high level, our algorithms follow a recursive-sampling approach. This is a standard idea in clustering and has been widely used, including in [Har-Peled\& Mazumdar (2004)], [Mettu & Plaxton (2002)], as well as prior work in weak-strong models such as [Galhotra et al. (2014)], [Bateni et al. (2024)], [Braverman et al. (2025)], and [Raychaudhury et al. (2025)]. However, aside from this broad similarity, our algorithms differ considerably. Working entirely with the weak quadruplet oracle under persistent probabilistic noise brings in several challenges absent from prior work and requires several new technical ideas. The ability to repeat queries or access even a small amount of distance information would simplify many steps substantially. We elaborate below.
>
> We begin with [Har-Peled \& Mazumdar (2004)]. At a high level, their algorithm in each round samples a set $S$ of $O(k \cdot \mathsf{polylog}(n))$ centers, finds for every point in $V$ its nearest neighbor in $S$, orders all points by this nearest-neighbor distance, and then partitions them into $O(\log n)$ buckets based on this distance. They then identify the rightmost \emph{heavy} bucket and remove all points to its left. Prior work in the RM-model, such as [Galhotra et al. (2014)] and [Raychaudhury et al. (2025)], shows how to adapt this framework while keeping the distance-query complexity small. However, both the approximate nearest-neighbor and the partitioning steps fundamentally rely on distance information; only the ordering step can be done using the noisy oracle. Hence, in the R-model these two steps are not feasible and we must proceed differently.
>
> We introduce several new ideas. We show that although we cannot find nearest neighbors for all points of $V$ with respect to $S$, we can do so for most points. To achieve this, we introduce a new technique based on \emph{kernel sets} and \emph{guard sets}, both identified from a second sample. Guard sets allow us to filter out the vertices for which nearest-neighbor identification is unreliable; kernel sets then allow us to find nearest neighbors for the remaining vertices. For this latter step, we show that kernel sets can be used to obtain a stronger form of approximate ordering than what is typically achievable using quadruplet oracles under probabilistic noise.
> To bypass the partitioning difficulty, we build on a known observation that shows that using a slightly larger sample $S$, one can avoid bucketing entirely and instead eliminate a constant fraction of vertices after ordering them according to their nearest-neighbor distance to $S$. We cannot apply this observation directly because of how our approximate nearest-neighbor routine works, but with additional care we show how to make everything work. In the bounded doubling dimension setting, we provide improved query complexity bounds, but this requires several additional ideas as we cannot replicate the filtering and nearest-neighbor strategies for the general case.
>
> Q2. This is a great suggestion. We added such a table in the revision.
>
> Q3. It is known from Galhotra et al. (2024) that for $n$ objects, no algorithm can compute a $o(n)$-factor approximation for $k$-median or $k$-means even in Euclidean space and even with unlimited quadruplet queries. By “sublinear” we were referring to this approximation bound. We clarified this in the revised submission.

---

### Official Review · Reviewer_6phy · 2025-10-29

**Soundness:** 3
**Presentation:** 3
**Contribution:** 3
**Rating:** 8
**Confidence:** 3

**Summary:**

This paper studies the metric k-clustering problem when only a weak comparison-based oracle (noisy oracle) is available. It introduces new representative centers construction method (with size $O(k poly\log n)$) together with a mapping strategy. Using only quadruplet queries, this paper gives randomized algorithms that achieve: 1) in general metrics, an O(1)-Coreset+ (an extended notion of coreset with tighter guarantees) of size $O(kpoly\log n)$ with $O(nk poly\log  n)$ queries; 2) in bounded-doubling metrics, the same-size Coreset+ construction with $O((n+k^2) poly\log n)$ queries; 3) for $k$-median/means objectives in doubling metrics, an $\epsilon$-Coreset+ under the same query bound. The key techniques include probabilistic sorting under persistent noise with $O(logn)$ dislocation and an adversarial-oracle emulation using “kernel/guard” samples to produce 4-sorted orders and enable ANN selection. Overall, the framework shows how low-cost and noisy oracles can be integrated into scalable clustering algorithms designs.

**Strengths:**

One of the paper’s key strengths is that it eliminates distance oracles entirely, which provides a clear breakthrough from RM-model work (e.g., Raychaudhury et al., 2025). This makes the proposed methods more practical in settings where exact pairwise distances are costly to compute, especially for high-dimensional or graph-based data settings. Building on the Coreset+ framework and using only quadruplet queries, the proposed method reduces computational complexities. It achieves query complexities of $O(nk poly\log n)$ in general metrics and $O((n+k^2) poly\log n)$ in doubling metrics, which are competitive with prior results while avoiding any distance computations.

Additionally, the algorithms are proved to remain within logarithmic factors of optimal bounds. Their techniques used, including probabilistic sorting and adversarial-oracle emulation, are new and interesting, which could have broader applications beyond clustering.

**Weaknesses:**

The weakness can be summarized as follows.

1.Limited Experiments. While the authors present results on a synthetic 2D dataset, the empirical evaluation is limited to a single dataset with fixed parameters. There is no exploration of how the algorithm performs with varying parameters, dimensions, and noise levels. The paper would benefit from additional experiments with different datasets, noise rates, and metric spaces.

2. Assumptions on noise. The paper assumes that noise across edge pairs is persistent and independent, which may not hold in practical settings. Some discussion of how to handle noise inconsistencies would improve the robustness of the proposed framework.

3.Practical Efficiency: The constants involved in the query complexity could be high for large-scale datasets. More discussion on how to choose parameters (e.g., m_win, kernel sizes) for practical applications would strengthen the paper's relevance to practitioners.

**Questions:**

1. What are the tight query lower bounds for this problem under weak (noisy quadruplet) oracles? Whether the upper bounds provided in this paper can match these limits (up to logarithmic factors) for k-median/means, and if not, where is the gap?

2. Since the paper focuses on general metrics, can the algorithms be further extended to low-dimensional Euclidean spaces—for example, achieving faster query complexities or stronger accuracy guarantees by leveraging geometric structure?

---

> ### Author Response · Authors · 2025-11-22
>
> We thank the reviewer for the thoughtful comments.
>
> W1. We have run additional experiments on higher-dimensional real datasets (and with different noise rates), and our results confirm the pattern observed in the synthetic experiments, namely, the k-means cost of our algorithm is consistently lower than that of the baseline. We included these results in Section 4 of the revised submission.
>
> W2. This is an excellent question. In theory, it remains an open problem whether one can obtain provable guarantees without the independence assumption. This assumption is also made in [Raychaudhury et al. (2025]), [Galhotra et al. (2024)], [Addanki et al. (2021)], [Bateni et al. (2024)], and [Braverman et al. (2025a)]. It may be possible to design approximation algorithms whose number of calls to the quadruplet oracle depends on the degree of dependency, but such results are beyond the scope of the current submission. In [Addanki et al. (2021)], the authors observed that the outcomes are robust even when the errors are not independent; they simulated the quadruplet oracle in the R-model using a random forest, and the empirical results were very close to the expected theoretical behavior under the independence assumption.
>
> W3. In the synthetic experiments, as well as in the new experiments on real datasets that we included in the revision, we used the following parameter settings: kernel size is roughly $\log(n)$, guard size is roughly $\log(n)/2$, and $m_{\text{win}}$ is roughly = $\log(n)/4$. Across all datasets tested under these settings, the empirical behavior was consistent with our theoretical guarantees.
>
> Q1. It is known from [Galhotra et al. (2014)] that for $k$-median and $k$-means clustering in the R-model, no $o(n)$-approximation algorithm exists, even in Euclidean space and even when unlimited quadruplet queries are allowed. In this submission, we match this lower bound up to an $O(\mathsf{polylog}(n))$ factor in the number of centers returned and an $O(1)$ factor in the clustering cost. In particular, we return a set of $O(k \cdot \mathsf{polylog}(n))$ centers whose clustering cost is $O(1)$ times the cost of the optimal $k$-median/$k$-means clustering.
>
> Q2. This is indeed the case for our Alg-D, as discussed in the paragraph “Overview of ALG-D” in Section 3. Please also refer to the second question on page 2 of the Introduction. While our general algorithm Alg-G runs in arbitrary metric spaces, Alg-D (and Alg-DI) is an improved algorithm specifically for metric spaces with bounded doubling dimension. A Euclidean space of constant dimension is a metric space with a bounded (constant) doubling dimension. Whereas Alg-G requires $O(n \cdot k \cdot \mathsf{polylog}(n))$ queries for general metrics, Alg-D requires only $O((k^2 + n) \cdot \mathsf{polylog}(n))$ queries in bounded doubling metric spaces (including constant-dimensional Euclidean spaces). We further highlighted this result for Euclidean spaces in the Introduction of the revised submission.

---

> ### Comment · Reviewer_6phy · 2025-11-26
> **Response to the Authors**
>
> Thanks for the clarifications. I prefer to keep my initial evaluations.

---

### Official Review · Reviewer_DLzD · 2025-10-30

**Soundness:** 3
**Presentation:** 4
**Contribution:** 3
**Rating:** 8
**Confidence:** 4

**Summary:**

The paper studies metric k-clustering problems in the so-called Rank-model (R-model). In this model, we only have access to a noisy quadrupled oracle, i.e. an oracle that returns a correct answer to the question “is point A closer to B, or is C closer to D?” with probability phi. Since it it known that it is impossible to get any sublinear approximation for k-median and k-means problems in this setting, the paper presents algorithms that computes O(k polylog n) centers (and the corresponding mapping of the other points), whose corresponding k-clustering cost is a constant approximation of the optimal k-clustering cost, using only O(nk poly log n) oracle calls.

The paper further presents an improved algorithm with the same number of centers, but with a run-time of O((n+k^2)polylog n) for metrics whose doubling dimension is bounded. Then, an even better (1+epsilon) approximation algorithm is possible to obtain for k-median and k-means problems in these metrics. Finally, using these obtained centers from each of the above algorithms, one can also easily compute coresets for these clustering problems.

** Technical Overview **

The generic algorithm follows a standard clustering approach by Mettu-Plaxton: we start by sampling a set S of O(k polylog n) points from the input set, and then order the remaining points by their distance to S. We then remove the closer half of these points, and recourse on the remaining ones. Of course, the biggest challenge is to find this ordering of the distances to S, since we don’t have an exact distance oracle.

The way to deal with this difficulty is to approximately sort the points in each step. Namely, if one uses a stronger notion of oracle, the so-called quadruplet oracle with adversarial noise, Raychaudhury (2025) showed that we can approximately sort these points by their distance from S. This oracle is similar to our weak oracle, but it returns the correct solution if the points are not too close, while it can return a false answer if the points are close. However, to be able to emulate this stronger oracle with our weaker quadruplet oracle, certain conditions must hold. Therefore, in all three algorithms, our goal is to find these conditions, and then exploit the above result.

In the general algorithm ALG-G, the algorithm first finds sets Kernel and Guard for each s in S. Guard points filter out the remaining input points that are too close to S, while Kernel points help us emulate the quadruplet oracle with adversarial noise.

In algorithm ALG-D we cannot afford to compute sets Kernel and Guard for each s in S. Instead, we first partition points in S into smaller parts, such that no two points in the same class are too close. With a slightly different filtering approach compared to ALG-G, we proceed in a similar manner to obtain the desired centers, now with respect to these parts. Finally, centers returned by ALG-D can be used to obtain clustering with better approximation by growing concentric balls around each of the centers, and exploiting the fact that the doubling dimension is bounded.

**Strengths:**

The paper is well-written and easy to follow.
The weak-oracle model is well motivated, and oracle-based modes in general have gained significant attention in the last few years.

**Weaknesses:**

While it is clear why ALG-DI required bounded doubling dimension, it is not entirely clear why ALG-D cannot be used in the general setting
Minor technical comments (see Detailed technical overview)

**Questions:**

What is the (intuitive) explanation why ALG-D fails in the general setting, i.e. where is the doubling dimension exploited?

** Detailed technical comments **

Line 24: from constant to epsilon -> to 1+epsilon?
Page 3: definition of O(1)- and epsilon-coreset not consistent, the second condition missing in line 120?
Lemma 2.1: |E|=m missing
Bounds on phi not consistent, is phi < 1/4 or < 1/2?

---

> ### Author Response · Authors · 2025-11-22
>
> We would like to thank the reviewer for the thoughtful and constructive comments.
>
> ALG-D: This is an excellent point. Indeed, we did not clearly explain in the overview why ALG-D requires a bounded doubling dimension. As we noted in the overview, at a high level, in each round ALG-D samples a subset of points, partitions the points through a careful process, and then constructs a nearest-neighbor data structure based on a result of Raychaudhury et al. (2025) on each subset. This data structure requires the doubling dimension to be bounded. We forgot to emphasize this point in the submission but we clarify this in the revision. In fact, the data structure of [Raychaudhury et al. (2025)] is the only part that requires the doubling dimension to be bounded; the rest of the steps remain valid in general metric spaces.
>
> Typos: Thank you also for your careful reading. We fixed the typos in the revision.
>
> Coresets: The definitions of $O(1)$-coresets and $\varepsilon$-coresets that we use are consistent with those in [Galhotra et al. (2024)] and [Har-Peled and Mazumdar (2004)], respectively. These two notions are fundamentally different, and the distinction is not merely the parameters $O(1)$ and $\varepsilon$. An $O(1)$-coreset is a weaker notion: it is a weighted set $A$ such that any $k$-clustering $\gamma$-approximation on $A$ yields an $O(\gamma)$-approximation for the original set $V$. In contrast, an $\varepsilon$-coreset is a weighted set that approximates the $k$-clustering cost of any set of $k$ centers within a $(1+\varepsilon)$ factor of the true cost on $V$. This is a stronger definition, which we only use in our final result. To clearly distinguish these notions, in the revised submission, we adopted the terminology $O(1)$-coreset and $(k,\varepsilon)$-coreset, following [Galhotra et al. (2024)], who also use both definitions.
>
> Definition of $m$: In Lemma 2.1, $m$ denotes the size of $X$. To avoid confusion, we have removed this notation and now refer to $|X|$ directly.
>
> Bound on $\phi$: Thank you for pointing this out. All our algorithms work for any probabilistic noise $\phi<1/4$. This bound comes from [Geissmann et al. (2025)], whose algorithm we use to approximately order distances with the noisy oracle. While it is true that some of our algorithms (for example Alg-G) can be extended to work for any $\phi<1/2$, for the sake of simplicity,  we decided to present the slightly weaker result. We ensured that the bound stated is consistently $1/4$ in the revision.

---

### Official Review · Reviewer_y7gL · 2025-10-31

**Soundness:** 4
**Presentation:** 4
**Contribution:** 4
**Rating:** 8
**Confidence:** 4

**Summary:**

The paper studied coreset algorithms with a noisy quadruplet distance oracle. Here, we are given $n$ data points in a metric space, and the distance metric $d$ is hidden from the algorithm. Instead, the algorithm could only make *comparison queries* in the form of ‘whether the distance $d(x,y)$ is smaller than the distance $d(u,v)$’. The oracle answers correctly with probability $1-p$ for some $p\in(0,1/4)$, and the answer could be adversarial with probability $p$. The noise is persistent, meaning that if the oracle returns a wrong answer, we cannot query multiple times to get the correct answer. We want to understand what we can do for $k$-clustering with information from the noisy quadruplet oracle answers.

The motivation for this model is from a recent line of work using weak and strong oracle models, where the information obtained by the weak oracle is usually deemed as ‘cheap but inaccurate’ and the strong oracle simply returns the correct answer, but it is considered expensive to query. In many cases, the presence of a strong oracle is necessary, as shown by Galhotra, Raychaudhury, and Sintos [PODS’24], that we cannot hope to obtain $O(1)$-approximation for $k$-clustering with only the weak quadruplet oracle.

This paper, in contrast, shows that it is possible to construct *coresets* of size $k \text{polylog}(n)$ using only the weak oracle. The catch here is that we might be able to output a subset of points that contains the information for $O(1)$-approximation $k$-clustering *without* knowing the actual clustering. On this front, the paper obtained an algorithm that achieves $O(1)$-approximation coreset of size $k \text{polylog}(n)$ using only $nk \text{polylog}(n)$ weak oracle queries. Furthermore, when the doubling dimension of the metric is bounded, the number of queries could be further decreased to $(n+k^2)\text{polylog}(n)$.

**Techniques.** The starting point of the techniques is quite simple: we know that if we sample $k \text{polylog}(n)$ points in each iteration, filter out a tiny constant fraction of the vertex pairs with the longest distances, then the rest of the points would preserve the distances by an $O(1)$ factor. Therefore, if we simply want a coreset, we could just sample recursively and order the distances from the quadruplet oracle with some noisy sorting algorithm. The process terminates in $O(\log{n})$ rounds, so we have coresets of size $k \text{polylog}(n)$. However, if we want to generate a mapping between the points and their coreset points, then we will need to be more careful. In particular, the distances sorted by the noisy algorithm are not trustworthy, so the paper designed a more careful’’guard’’ set that basically serves as ‘distance witnesses’ to filter out points that are too close to the sampled points. For the rest of the points, since we could tolerate the radius of the kernel set distance, we could aggregate the information to compute an approximate neighbor for $v$ in the sampled set. This constitutes the main idea used in the paper.

**Strengths:**

Overall, I like this paper. It follows the recent line of work for weak-strong oracle models. However, different from existing results that almost exclusively showed the necessity of the strong oracle, this paper is the first to show that we could do something without the strong oracle at all (as far as I know). I think this is a nice conceptual contribution. The paper is well-written, and although I did not get the time to check all the steps, the technical overview is easy to follow. Therefore, I would want to see the paper accepted into the conference.

**Weaknesses:**

I do not see any major weakness in the paper. One thing I want the author to emphasize is that one can know the coreset without knowing the clustering, since I was confused for a moment about whether there is a contradiction with GRS [PODS’24].

Some of the technical overview is a bit wordy and dense with math. I understand it’s a bit hard to present it in a cleaner manner. Maybe you can add some figures for the guard and kernel sets, plus the filtering process. I believe that’ll help readers understand.

**Questions:**

I have some slightly technical questions to help me understand the results better:
- If we do not care about the mapping between points and the coreset, but we are still given the noisy oracle (as opposed to the perfect one), does it make the situation drastically easier? I believe it is the case since the dislocation error is at most $O(\log{n})$, and the recursive process that eliminates O(1) fraction from the sampled sets is going to succeed with high probability. Is this correct, or did I miss anything?
- For the points that do not get assigned to any sampled coreset point, e.g., the points in the kernel, do they simply survive for the next round of sampling? I guess this is the reason you want $|Kernel|<= V_i/\text{polylog}(n)$, but I want to confirm.
- Also, how large is the exponent on the logarithm terms? Do you have an upper bound?
- Finally, line 264 says ‘For simplicity, we focus on the k-median objective (p = 1), but our approach extends naturally to $p>1$’ —-- I don’t really buy this. $k$-median costs have some nice linear properties that other clustering objectives might not have. Do you want to use generalized triangle inequality? You’ll lose a factor of $2^p$, though. Then, you’ll need to say that you assume $p=O(1)$.

---

> ### Author Response · Authors · 2025-11-22
>
> We would like to thank the reviewer for the insightful review.
>
> We added a figure that depicts the kernel and the guard sets, and explains the filtering process.
>
> Q1. Your observation is indeed correct. In general metric spaces, if we do not care about the mapping, then the problem does become simpler. However, the crux of the problem lies in constructing the mapping, as without it one cannot actually “cluster” the data. This is exactly why we emphasize that the goal of our algorithms is to compute a coreset+. We also note that obtaining the improved query complexity for low-dimensional settings remains challenging, even without the mapping.
>
> Q2. The points in the kernel (and in the guard) are always removed at the end of a round and become part of the coreset. Please notice that the algorithm recurses on $V_{i+1} = V_i \setminus (V_i’’ \cup S_i^{(1)} \cup S_i^{(2)})$, and all kernels and guards are subsets of $S_i^{(2)}$. If $|V_i| = O(k \cdot \log^3 n)$, or equivalently $|V_i| = O(|Kernel| \cdot \log n)$, then we can add $V_i$ to the final set of centers $C$.
>
> Q3. We made no attempt to optimize the log factors. Nonetheless, they are not too big, and can be safely upper-bounded by 8 in all cases.
>
> Q4. Thank you for your careful reading! Indeed, we forgot to add that p = $O(1)$.

---

> > ### Comment · Reviewer_y7gL · 2025-11-22
> >
> > I have read your comments and believe that my questions have been answered. Thanks! I maintain my original positive evaluation, and I'll be happy to see the paper gets accepted.

---

### Official Review · Reviewer_8Vye · 2025-11-01

**Soundness:** 4
**Presentation:** 3
**Contribution:** 4
**Rating:** 6
**Confidence:** 4

**Summary:**

The paper studies metric $k$-clustering when instead of distances an oracle is provided that given 2 pairs of points $(a,b)$ and $(c,d)$ returns if $a$ is closer to $b$ or $c$ is closer to $d$. Previous works has establish that with only access to oracle and no distance information, no constant approximation algorithm exists with $k$ centers. This work instead considers constructing coresets, and give and algorithm that outputs a $O(1)$ error coreset with $O(k\text{poly}\log{n})$ points and $O(nk\text{poly}\log{n})$ oracle queries. If the underlying metric has bounded doubling dimension, the query complexity improves to $O((n+k^2)\text{poly}\log{n})$, and $(1+\varepsilon)$-coreset for $k-$means and $k-$median.

**Strengths:**

The problem setting is very natural, for cases where distance computation might be hard or expensivem, performing comparisions using machine learning models might be efficient. The proofs are clear to the best of my knowledge. This work removes the requirement of distance oracle and provides coreset construction with only the oracle queries. The results obtained are near-optimal.

**Weaknesses:**

The experimental setting is very limited, with them being run on only one synthetic dataset. But the algorithmic contributions and results obtained are non-trivial and significant contribution so this is not really a major concern for me. Maybe a minor typo - probsort (in Lemma 2.1) requires noise parameter $\leq 1/4$ but appendix A says $\leq 1/2$.

**Questions:**

N/A.

---

> ### Author Response · Authors · 2025-11-22
>
> We would like to thank the reviewer for the constructive comments.
>
> Experiments:
> We have run additional experiments on higher-dimensional real datasets (and with different noise rates), and our results confirm the pattern observed in the synthetic experiments, namely, the k-means cost of our algorithm is consistently lower than that of the baseline. We included these results in Section 4 of the revised submission.
>
> Probabilistic noise:
> Thank you for your careful reading. All our algorithms work for any probabilistic noise $\phi<1/4$. This bound comes from [Geissmann et al. (2025)], whose algorithm we use to approximately order distances with the noisy oracle. While it is true that some of our algorithms (for example Alg-G) can be extended to work for any $\phi<1/2$, for the sake of simplicity,  we decided to present the slightly weaker result. We ensured that the bound stated is consistently $1/4$ in the revision.

---

> > ### Comment · Reviewer_8Vye · 2025-11-26
> > **Reponse to authors**
> >
> > I thank the authors for their response. I maintain my positive evaluation of the work.

---

### Author Response · Authors · 2025-11-22

We thank the reviewers for their thoughtful and constructive feedback. We have uploaded a revised version of our submission, and for the reviewers’ convenience, all changes are highlighted in blue.

The major updates are as follows:

(i) We added Table 1 to the Introduction, which compares our algorithms with existing clustering algorithms in the R- and RM-models.

(ii) We added Figure 1 to Section 3 to improve the presentation of the algorithm. The figure illustrates the kernel and guard sets and clarifies the filtering process.

(iii) We conducted additional experiments on higher-dimensional real-world datasets, varying the probabilistic noise $\varphi$ and the number of centers $k$. The results confirm the trends observed in the synthetic experiments: the $k$-means cost of our algorithm is consistently lower than that of the baseline. These results are now included in Section 4.

(iv) We added a new lower bound in Appendix A showing that any $o(n)$-approximation algorithm for $k$-median/means clustering in the R-model must return at least $2k-1$ centers.

---

### Meta-Review · Area_Chair_DvcP · 2025-12-23

**Summary:**

The paper provides the first algorithmic $k$-clustering results in the purely weak oracle model (R-model) by relaxing the constraint on the number of output centers. It demonstrates that while finding exactly $k$ centers is impossible, constructing a representative coreset of size $\tilde{O}(k)$ is feasible with near-optimal query complexity. The work introduces novel mechanisms (kernels/guards) to handle persistent probabilistic noise in comparisons without resorting to exact distance queries.

Strengths:

- The paper successfully removes the dependency on "strong" (exact distance) oracles, which was a limitation in prior work (RM-model).
- The adaptation of recursive sampling techniques using only relative comparisons is non-trivial.

Weaknesses:
- Reviewers noted the experiments were initially limited to synthetic data. The authors added real-world datasets (Adult, Credit Card) in the revision, though the evaluation remains relatively small-scale
- The theoretical guarantees rely on the assumption that oracle noise is independent across queries

Overall, the paper is a nice theoretical contribution and it would be a nice addition to the ICLR program

**Reviewer Concerns:**

All the concerns have been already addressed in the initial rebuttal

**Reviewer Scores:**

I do not think that the scores would have changed more if there was more time for discussion.

---

### Decision · Program_Chairs · 2026-01-26

Accept (Poster)